# Diagnosing Aerosol-Meteorological Interactions on Snow within Earth System Models: A Proof-of-Concept Study over High Mountain Asia

Chayan Roychoudhury[1], Cenlin He[2], Rajesh Kumar[2], and Avelino F. Arellano Jr.[1]

[1]Department of Hydrology and Atmospheric Sciences, University of Arizona, Tucson, AZ, USA
[2]Research Applications Laboratory, NSF National Center for Atmospheric Research, Boulder, CO, USA

**Correspondence:** Chayan Roychoudhury (croychoudhury@arizona.edu)

**Abstract.** Snowmelt in the Third Pole, or High Mountain Asia (HMA), serves as a vital water source for 30% of the world's population and is strongly influenced by interactions between aerosols and meteorology. However, understanding these interactions remains uncertain due to their complexity and limitations in existing approaches using model sensitivity and process-denial experiments. In addition, these interactions are insufficiently represented in current climate models. Equally ambiguous is the impact of these interactions on snow processes in the context of climate change. Here we use network theory, a graphical approach that maps the relationships between variables as interconnected nodes, to identify key variables that influence snowmelt processes. We focus on the late snowmelt season (May-July) using daily data (from 2003-2019) from satellite observations and reanalyses. We combine statistical and machine learning methods to highlight the underappreciated relevance of coupled processes between aerosols and meteorology on snow, as well as the inconsistent representation of aerosol-meteorology interactions on snow within major reanalyses. These inconsistencies reflect fundamental differences in model design. In particular, we identify underrepresented dust interactions with near-surface temperature and large-scale circulation and gaps in cloud cover interactions, especially in the least coupled reanalysis. Carbonaceous aerosols and large-scale circulation emerge as main drivers of aerosol-meteorology onto snow interactions, highlighting their relevance in Earth system models (ESMs) for the accurate assessment of water availability in developing economies. These insights point to the degree of complexity of these interactions and their relative strength of representation across ESMs. The proposed framework can be extended to help diagnose other complex Earth system processes and complement conventional feedback separation methods. This has broader implications for the future development of coupled models to improve Earth system predictability.

## 1 Introduction

The rapid acceleration of glacial snowmelt in recent decades has critically impacted the freshwater resources that serve the livelihood of regions downstream of the glaciers in High Mountain Asia (HMA), often referred to as the Third Pole that contains the world's largest reservoir of glaciers and snow (~1% of the Earth's surface area) outside of the Earth's polar ice sheets (Kraaijenbrink et al., 2021; Yao et al., 2019; UNEP, 2022). These snowmelt trends reflect the susceptibility of HMA to climate change and the complex interplay between the land and atmosphere, which has cascading effects on snowmelt and

water resources downstream for approximately 2 billion people (Mudryk et al., 2020; Barnett et al., 2005). Snow cover fraction (SCF), which is one of the ways we can represent snowmelt, is an essential climate variable. SCF modulates the surface energy balance and atmospheric circulation (Cohen and Entekhabi, 2001; Cerveny and Balling Jr., 1992). The high albedo of snow leads to feedback mechanisms where the warming-induced snowmelt exposes the underlying darker surfaces, absorbing more solar radiation and further accelerating warming and snowmelt. These cryospheric changes influence the Earth's radiation budget and contribute to the broader Earth system feedbacks that drive climate change (Flanner et al., 2011; Robock, 1983; Hileman, 1992).

Past studies on SCF variability over HMA have primarily focused on meteorological factors (e.g., ambient temperature, precipitation) and topography, often overlooking anthropogenic emissions in the vicinity of glaciers (Bonekamp et al., 2021; Sorg et al., 2012; Pepin et al., 2015; Singh et al., 2022). Light absorbing particles (LAPs, viz. black and brown carbon (BC, BrC), and dust), a key component of these emissions, have a considerable impact on snow cover through deposition with efficacy comparable to greenhouse gases (Warren and Wiscombe, 1980; Qian et al., 2015; Kang et al., 2020; Shindell and Faluvegi, 2009; Sarangi et al., 2020; Brown et al., 2022; Wang et al., 2023). The interactions between meteorology and aerosols (mostly but not limited to LAPs) onto snow, defined as aerosol-meteorology interactions at the snow interface (AMI) affect (a) the Earth's radiation budget and atmospheric thermodynamics; (b) cloud microphysical properties, precipitation rate, and type; (c) snow albedo through deposition and snow darkening; (d) snow physical properties (e.g. specific surface area) that amplify the snow albedo feedback; and (e) the hydrological cycle through increased ground temperatures, decreased snow water equivalent, shortened snow cover duration, and earlier runoff (ram; He, 2022; Andreae and Rosenfeld, 2008; Painter et al., 2007; Lohmann and Feichter, 2005; Flanner et al., 2007; Huang et al., 2022; Lohmann, 2017; Borys et al., 2000; Oaida et al., 2015). Previous studies rarely incorporate this full spectrum of variables and pathways in hydrological analyses (Wu et al., 2018). The complexity is further compounded by the diverse sources of these LAPs: BC and BrC originate predominantly from anthropogenic sources, while dust comes from natural sources in desert regions in the vicinity of HMA (Shi et al., 2019). The spatial heterogeneity in HMA's glaciers and the non-linear interaction between these processes can either intensify or buffer the response of the climate system, confounding the net effect of AMI on the cryosphere and potentially leading to the misattribution of the relevant drivers to snowmelt (Sakai and Fujita, 2017; Ragettli et al., 2016; Kapnick et al., 2014; Bonekamp et al., 2019; Sand et al., 2020; Stevens and Feingold, 2009; Michibata et al., 2020; Gautam et al., 2013; Rahimi et al., 2020). For instance, recent studies place contrasting degrees of importance on the types of LAPs and their radiative impact on snow. While BC has historically dominated such studies, emerging research indicates dust and BrC have greater significance than previously recognized (Tuccella et al., 2021).

Understanding these interacting processes is crucial for accurate climate change predictions, particularly regarding freshwater availability in Third Pole regions like HMA (Zhang et al., 2019). A systems-science approach to these non-linear processes conceptualizes the Earth's climate as a self-regulating system of organized complexity with interconnected feedbacks among its sub-components (Wood Jr, 1988; Steffen et al., 2020; Kump et al., 2010). Ripple et al. (2023) identified approximately 41 of the most significant climate interactions/feedbacks driving climate change, seven of which are yet uncertain, the major source of uncertainty being atmospheric chemistry (Heinze et al., 2019; Pathak et al., 2023). Current state-of-the-art ESMs have only

recently been able to incorporate some degree of atmospheric composition feedbacks into coupled atmosphere-ocean models (van Vuuren et al., 2012; Randall et al., 2018; Giorgi and Gao, 2018). As models incorporate more processes and interactions, they exhibit an increased uncertainty in projections of various climate variables such as surface temperature and precipitation (Meehl et al., 2007; Stainforth et al., 2005; Pathak et al., 2023), which highlight the fundamental gaps in current ESMs to fully simulate these coupled Earth system processes, particularly at regional scales. Therefore, rigorous evaluation and assessment of ESMs and coupled regional models is essential, in addition to improved quantification and parameterization of these diverse feedback mechanisms.

Challenges in constraining these non-linear interactions in these models arise from a variety of factors: the sparsity in long-term continuous observations, theoretical uncertainties in parameterization and coupling, spatio-temporal heterogeneity in the processes, as well as the magnitude and direction of these feedbacks. Researchers have traditionally addressed these uncertainties through multiple approaches: 1) perturbing model parameters and sensitivity analysis 2) selective withholding of modeled parameters and comparing their relative impacts, 3) using observations and data-assimilation (Schneider et al., 2017), and 4) assessing emergent constraints across multiple models, (Heinze et al., 2019; Soden et al., 2008; Zhou et al., 2019; Moch et al., 2022; Gettelman, 2015; Barthlott et al., 2022; Archer-Nicholls et al., 2016; Usha et al., 2020; Stein and Alpert, 1993). However, the computational demands of these methods necessitate alternative approaches. Simpler statistical techniques ranging from regression to emulators offer more accessible insights albeit with limitations (Carslaw et al., 2013; Johnson et al., 2015; Lee et al., 2013; Xiao et al., 2023; Wall et al., 2022; Gregory et al., 2004). Artificial intelligence methods to emulate complex systems provide viable alternatives (Reichstein et al., 2019; Shin et al., 2022; Schneider et al., 2017; Irrgang et al., 2021), despite being perceived as black boxes that introduce additional complexity and biases (Castelvecchi, 2016; Lipton, 2018). Following the systems-science paradigm, Harte (2002) proposes integrating Newtonian (top-down, system-wide analysis) and Darwinian (bottom-up, process-level analysis) perspectives to better understand complex Earth system processes and potentially constrain some of their uncertainties. Harte's approach has been adapted for aerosol-cloud interactions (ACI) to constrain ACI-related feedbacks through combined process modeling and observations (Feingold et al., 2016). In this study, we apply a similar methodology to diagnose these feedbacks at the snow interface in HMA, where the system-wide complexity is reduced into second-order non-linear interactions between simpler sub-components. Our approach attempts to identify second-order feedbacks between atmospheric composition (particularly aerosols) and meteorology that influence the cryosphere interface (i.e., snow surface) over HMA. Examples of these second-order interactions can include (a) interactions between absorbing aerosols and geopotential height that can modify convection patterns, regional temperature, and precipitation, and (b) interactions between BC and near-surface temperature that reduce snow albedo through near-surface warming.

Our definition of AMI has an additional layer of complexity due to the cryosphere and the land-atmosphere interface. Unlike Feingold et al.'s approach, which explores ACI as a product of the sensitivities of different cloud properties to cloud radiative forcing, we examine the joint dependencies (pairwise or two predictors simultaneously) between aerosol and meteorological drivers of snowmelt through observations and reanalysis data. Previous studies have typically adopted either a Newtonian perspective, assessing snow-related sensitivity to aerosols (Usha et al., 2020; He et al., 2018), or a Darwinian perspective, attributing snow-related trends to dominant meteorological factors (Sorg et al., 2012; Pepin et al., 2015). The missing element is

the insight into the feedback/interacting processes between aerosols and meteorology on the snow interface, or the Darwinian-
Newtonian nexus, which we incorporate in this study. Fig. 1 illustrates our integrative approach. The Darwinian view identifies
individual aerosol and meteorological factors that couple among themselves and impact snow properties. The Newtonian view
examines the physics-based processes like aerosol deposition, snow property modification, accumulation, and ablation (melt) in
the system-wide atmosphere-cryosphere interface. We bridge these perspectives by considering individual predictors relevant
to snow processes (Darwinian) while highlighting specific interactions influencing snowmelt (Newtonian).

Our objective in this study is to demonstrate this system-science approach for AMI on snow over HMA. The lack of consis-
tent geophysical observations across HMA, driven by its remoteness and complex terrain, compels us to use reanalysis products
where observations already constrain model simulations with long-term records. ERA5/CAMS-EAC4 and MERRA-2 are two
of the most widely used global reanalyses that have different approaches to coupling interconnected processes (Hersbach et al.,
2020; Inness et al., 2019; Randles et al., 2017). In our previous work (Roychoudhury et al., 2022, hereafter R22), we attempted
to address AMI on snow by quantifying the importance of second-order interactions between aerosol and meteorological vari-
ables from ERA5/CAMS-EAC4 reanalysis to satellite-based MODIS SCF using multi-linear regression. We found that AMI,
particularly interactions related to carbonaceous aerosols, hold high importance for glacial regions with low snow cover fraction
(LSC) in HMA, particularly during the late snowmelt season (May-July). Recently, a new regional reanalysis from NSF NCAR
(https://himat.org/topic/matcha/) was developed to meet the objectives of NASA's High Mountain Asia 2 project, particularly
focusing on aerosol deposition on snow. This presents an opportunity to evaluate AMI on snow's representation across these
reanalyses, given the known differences in coupling within their model design. This paper is a proof-of-concept that extends
R22 by quantifying the importance of AMI on snow over HMA across three reanalyses (ECMWF, NASA, and NSF NCAR).
(see Sect. 2.1) using both statistical multi-linear regression and explainable machine learning (ML) approaches. We evaluate
this importance with both observed and modeled SCF to understand the differences between observable and modeled AMI on
snow. The aim is to gain insights into the model representation of AMI-related processes in each reanalysis, rather than discov-
ering new physical mechanisms or causal pathways within AMI. Our results show that AMI contributes at least 20% to SCF
variability, with absorbing aerosols and large-scale circulation emerging as dominant processes requiring improved representa-
tion in current reanalyses. Through network visualizations and joint distributions, we illustrate the varying degrees of coupled
parameterization across reanalyses, demonstrating that accurate attribution of Earth system phenomena in climate-vulnerable
regions depends significantly on how interactions are represented in coupled models.

The remainder of this paper is structured as follows. Sect. 2 describes our data sources and methods. We elaborate on the
regression framework and quantification of the importance of AMI. Sect. 3 presents our results on the model differences in
AMI through network visualizations and identifies associations that align with existing studies. Finally, Sect. 4 summarizes
our main findings, discusses the implications for Earth System predictability, and acknowledges limitations while suggesting
directions for future research.

## 2 Methods

### 2.1 Reanalyses

ERA5 and its land counterpart ERA5-Land are the most recent fifth-generation reanalysis datasets from ECMWF (European Centre for Medium-Range Weather Forecasts) available from 1979 to the present (Hersbach et al., 2020; Muñoz-Sabater et al., 2021). MERRA-2 is the most recent reanalysis from NASA GMAO (Global Modeling and Assimilation Office) with data available from 1980 to the present (Gelaro et al., 2017). Both datasets are observationally constrained by assimilating multiple satellites and in-situ data. It should be noted that snow cover assimilation in ERA5-Land is limited as data above an elevation of 1500 m is not assimilated, while most of the study domain lies above this threshold. An important difference between the two reanalysis frameworks is 1) the inclusion of online coupling between aerosol and radiation in MERRA-2, whereas in ECWMF, the radiation scheme uses an aerosol climatology instead; and 2) a separate reanalysis (CAMS-EAC4) for atmospheric composition from ECMWF exists, while aerosol products are already simulated within MERRA-2 (Inness et al., 2019; Randles et al., 2017). We hereafter refer to the ECMWF reanalysis, ERA5 with CAMS-EAC4 as ERA5/CAMS4. In addition to these publicly available reanalyses, we also use a recent, regional reanalysis product from NSF NCAR (National Center for Atmospheric Research) called MATCHA (Model for Atmospheric Transport and Chemistry in Asia) consisting of 16 years of hydrometeorological and aerosol fields over HMA, generated using the WRF-Chem v3.9.1 (Weather Research and Forecasting Model with Chemistry) coupled with the CLM-SNICAR model (Community Land Model – Snow Ice Coupled with Aerosol and Radiation) (Kumar et al., 2024; Flanner et al., 2021; Oleson et al., 2010; Skamarock et al., 2008). The model framework in MATCHA couples variables between aerosols, meteorology, and land in two ways 1) the Rapid Radiative Transfer Model for General Circulation Models (RRTMG) allows for online interaction between simulated aerosols and radiation and 2) the use of SNICAR with CLM is an additional component in MATCHA that modifies snow albedo due to deposition of LAPs (Archer-Nicholls et al., 2016; Mlawer et al., 1997; Kumar et al., 2014). Similar to the reanalyses from ECMWF and NASA, MATCHA is observationally constrained by daily assimilation of MODIS AOD and MOPITT CO products (acronyms explained in Appendix B) to constrain the concentration and deposition of LAPs in Asia. These reanalyses encompass different meteorological models and representations of aerosol processes that make them suitable candidates for understanding the representation of interactions in each model framework. The general characteristics of these datasets are available in Table S1.

ERA5 exhibits systematic wet and warm biases over Asia, with higher near-surface wind speeds (Sun, 2017; Gong et al., 2022; Wei et al., 2024). CAMS-EAC4 captures large-scale aerosol transport but underestimates total and speciated aerosol concentrations, particularly during high aerosol events, and overestimates BC. MERRA-2 generally simulates higher dust concentrations and better represents extreme aerosol events, but overestimates BC (Li et al., 2024; Ansari and Ramachandran, 2024; Gueymard and Yang, 2020; Xian et al., 2024). It is also important to note that while these reanalyses are invaluable for studying regions with sparse observations like HMA, their application in high-elevation regions has its challenges. Several studies in the region have pointed out elevation biases in surface temperature and its trends, as well as wind patterns arising from complex topography, varying vegetation cover, and coarser reanalyses that cannot resolve valley-scale terrain (Luo et al., 2019; Tang et al., 2022; Jentsch and Weidinger, 2022; Pepin and Seidel, 2005).

A total of 22 variables (6 aerosol and 15 meteorology-related) from the three reanalysis datasets, in addition to elevation, were selected as predictors that can potentially drive SCF (see Table S2 for the predictors). The meteorological variables, defined hereafter as MET include a) temperature (2-m temperature and skin temperature), b) cloud cover (total, low, mid, and high-level cloud cover fraction), c) dynamic circulation (mean sea level pressure, geopotential height at 500 hPa and 300 hPa, 10-m zonal and meridional winds), d) surface energy fluxes (surface sensible and latent heat), and e) moisture (2-m
specific humidity, daily accumulated total precipitation). The choice of these variables was guided by previous studies showing temperature, precipitation, surface energy fluxes, and cloud cover are important factors across snow variability studies (Duan and Wu, 2006; Ohmura et al., 1992; Shi et al., 2013; Södergren et al., 2018; Wang et al., 2015; Harder et al., 2017; Shi et al., 2011; Senf et al., 2021; Schlögl et al., 2018). The dynamic circulation variables are chosen considering the association of wind-driven processes and atmospheric teleconnections on SCF (Jiang et al., 2019; You et al., 2020). The variable groupings
here are meant to reflect the modules within the architecture of coupled models/ESMs that interact as sub-components, aligning with a systems-science framework.

    Aerosol variables, defined as AER hereafter, consist of aerosol optical depth (AOD) at 550 nm and surface mass mixing ratios. These are grouped by species: a) carbonaceous (hydrophilic and hydrophobic BC and organic matter, b) dust, c) sulphate, and d) others (sea-salt surface mixing ratio including total AOD at 550 nm). Note that MATCHA does not separate carbona-
175 ceous aerosols into hydrophobic and hydrophilic components and uses an internal mixing assumption of different aerosol species from emissions. Table S2 provides an overview of the variables used in the reanalyses.

    In addition to aerosol and meteorological variables, we used elevation (defined as ELEV) from the Global Multi-resolution Terrain Elevation Data (GMTED 2010) as a static predictor to represent topography and its related interactions (Pepin et al., 2015; Danielson and Gesch, 2011). Although snow hydrology is found to be sensitive to not only elevation but also other
topographical factors like aspect, slope, and shadowing effects, we use only elevation for this study as a common static predictor to represent topographical interactions across the three reanalyses (Hao et al., 2021).

## 2.2   Satellite Data

We use MODIS-based Level 3 daily satellite products with a horizontal resolution of $0.05^o$, namely snow cover fraction (SCF) from MOD10C1/MYD10C1 Collection 6.1, AOD at 550 nm from MODIS processed using the MAIAC algorithm
(MCD19A2CMG version 6.1), and land-surface temperature (LST) (MOD11C1/MYD11C1 Collection 6.1) (Hall and Riggs, 2021b, a; Lyapustin, 2023; Wan et al., 2015a, b). We chose to assess our understanding of snowmelt using SCF as 1) it is recognized as an essential climate variable, 2) shown to determine the strength of snow albedo feedback, and 3) shows higher sensitivity to snow albedo feedback than snow albedo in some studies (World Meteorological Organization (WMO), 2022; Qu and Hall, 2007; Fernandes et al., 2009). This choice was also influenced by SCF's broad applicability to various stakeholders
for hydroclimate studies in data-sparse regions (Crumley et al., 2020). MODIS LST contains daily data for both day and night, averaged to a daily estimate of LST. MODIS LST is used as a surrogate variable for skin temperature from each reanalysis (Jin and Dickinson, 2010). SCF and LST products from MODIS contain products from both satellites, Terra and Aqua, which were averaged to a single quantity for this study. We also use daily accumulated precipitation from IMERG (post-processed final

runs) with a spatial resolution of $0.1^o$ to represent precipitation over HMA (Huffman et al., 2014). The acronyms used here are listed in Appendix B.

## 2.3 Regridding the Data

The finer pixels of the predictors from both reanalysis and satellites were spatially averaged to $0.75^o$, considering that AER variables from CAMS-EAC4 are available only at $0.75^o$. Hourly to 3-hourly products from each reanalysis (ERA5/CAMS4, MERRA-2, and MATCHA) were averaged to daily data between the years 2003 and 2018. An exception is the daily accumulated precipitation from the three datasets, which was calculated by aggregating (summing) the hourly products into daily products.

## 2.4 Glacier Regions

A total of 6 glacier regions (GRs) are defined for HMA following the classification in Randolph Glacier Inventory version 6.0 (RGI Consortium, 2017). A total of 15 second-order glacier regions were aggregated into 6 major GRs for this study (Roychoudhury et al., 2022) (see Fig. S1). These GRs refer to the geographical extent of the snow-covered regions containing the individual glaciers. The geographical extent of the GRs over HMA is shown in Fig. S1a. GRs marked in red (blue) denote regions of high snow cover or HSC (low snow cover or LSC) and have been identified using the methodology described in R22. We specifically focus on the late snowmelt season, i.e., May-July across the years 2003-2018, when AMI is found to be significant in LSC (blue) regions (Roychoudhury et al., 2022). The spatio-temporal mean (standard deviation) across LSC regions during 2003-2018 is 2.4% - 5.5% (5.6% - 9.1%). HMA has an average altitude of 4 km, with a large number of the highest mountains and plateaus in the world across both the LSC and HSC regions. The region is typically arid, with humid summers due to the Asian monsoon. The vegetation type is mostly grasslands and forests, with vegetation greening mostly concentrated in LSC regions as well as foothills of HSC regions within the recent decades (Liu et al., 2022, 2021b; Maina et al., 2022).

## 2.5 Regression Framework to Estimate the Importance of AMI on Snow

We regress the target variable (daily SCF) on 22 predictors spanning aerosols (AER), meteorology (MET), and elevation (ELEV) following the equation,

$$Y^{s,t} = \overbrace{\sum_p \alpha_p X_p^{s,t}}^{\text{Term 1}} + \overbrace{\sum_{p,q} \alpha_{pq} X_p^{s,t} X_q^{s,t}}^{\text{Term 2}} + \overbrace{\alpha \mathbb{O}(X^{s,t})}^{\text{Term 3}} \qquad (1)$$

where $\alpha$ is the importance/sensitivity of the predictors ($X$) in regulating SCF ($Y$), $p$ and $q$ denote sets of different types of predictors (AER, MET, and ELEV) with the superscripts $s$ and $t$ denoting the spatio-temporal dependency of the quantities. Term 1 represents the linear sensitivity of SCF as a function of the AER, MET, and ELEV variables (the predictors $X$). Terms 2 and 3 introduce non-linear effects that account for interactions between different predictors. Term 2 focuses specifically on

product interactions influencing snow grouped as 1) aerosol-meteorology interactions (AMI); 2) aerosol-elevation interactions (AEI); 3) meteorology interactions (MMI); and 4) elevation-meteorology interactions (MEI), with AMI as the primary focus in this study. In contrast, Term 3 points toward higher-order unresolved processes extending beyond second-order product interactions in Term 2. We select daily products of the target and predictor variables over a $0.75^o$ by $0.75^o$ grid, grouped by six glacier regions (GR) and three months within the late snowmelt season (May-July).

The importance metric, $\alpha$, is derived with two distinct methods: (1) relative importance (RI), obtained from the multi-linear regression described in Sect. 2.6 includes the linear predictors and their second-order product terms (Terms 1 and 2 in Eq. 1). The estimated importance values ($\alpha$) are in percentages, and the sum for all terms in the regression equals 100%. (2) Shapley contribution (SHAPc) calculated from an ML model introduced in Sect. 2.7. It is important to note that while the multiple linear regression for RI is trained on both the original predictors and the product terms to account for interacting effects, the ML model is trained only on the individual predictors, as its built-in feature contribution algorithm (see Sect. 2.7) also accounts for the pairwise interactions, acting as a bulk measure of the importance, thus the three terms in Eq. 1 (Term 1 to 3). The importance values calculated from machine learning are normalized so that their total also equals 100%. Thus, both importance metrics are expressed as percentages that sum to 100%, making their magnitudes directly comparable. Each $\alpha$ value is inherently bivariate as it quantifies the sensitivity of snow cover fraction (SCF) to a given predictor in the presence of another predictor. Importance of AMI on snow can thus be interpreted as the impact of MET predictors on SCF in the presence of AER variables.

Our notion of importance parallels the chain rule representation (the Darwinian paradigm) of the extensively studied ACI in the context of cloud radiative forcing, with cloud fraction/cover as one of the dependencies. While the product of sensitivities in the chain rule formulation may not fully capture the non-linear feedback (interaction) between its dependencies, we can draw a direct link between the importance of aerosol-cloud cover interactions in our definition of AMI and ACI-cloud cover sensitivity from past studies (e.g., Feingold et al., 2016).

## 2.6 Relative Importance Analysis (RIA)

A multi-linear regression (MLR) model was used to regress daily SCF ($Y$) on a total of 253 predictor variables represented by the equation,

$$Y = \sum_{i=1}^{22} \alpha_i X_i + \sum_{i,j=1;j\neq i}^{231} \alpha_{ij} X_i X_j \tag{2}$$

$$= \sum_{i=1}^{22} \left( \alpha_i + \sum_{j\neq i}^{21} \alpha_{ij} X_j \right) X_i \tag{3}$$

where $N(=22)$ is the original number of predictors (see Fig. 1 and Table S2 for the 22 predictors: six aerosol, 15 meteorological, and an elevation variable) representing the main effects, in addition to $\left( \binom{N}{2} = 231 \right)$ non-linear interaction terms defined as product terms between these predictors (excluding square terms), thus leading to 253 (= 231 + 22) predictors in total. We

explicitly define second-degree interaction terms in the MLR model (only non-square terms) shown in Eq. (2) to represent the non-linear sensitivities of our predictors to the SCF variability for each GR and each month in the late snowmelt season. The interaction terms belong to five groups, namely: 1) AER-AER, 2) AER-MET, 3) AER-ELEV, 4) MET-ELEV, and 5) MET-MET. Eq. (3) offers us an alternate understanding of such a pairwise interaction, where the dependence ($\alpha$) on a predictor $X_i$ is not a constant, but dependent on a second predictor ($X_j$). AMI on snow is defined herein as the sum of $\alpha$ (the importance on modulating SCF) for each predictor in the groups AER-AER and AER-MET, along with the main predictors from AER. We considered AER-AER to capture the snowmelt response to the bulk effect of aerosols in the presence of meteorology.

We estimate the importance ($\alpha$) of the main and interaction terms using relative importance analysis (RIA) that overcomes the issue of correlated predictors (which is very likely in our case) (Tonidandel and LeBreton, 2011). In line with our definition of importance, RI quantifies the impact of each predictor (both main and interaction effects) on SCF through fractional contribution to the total explained variance ($R^2$). Consequently, RI values sum up to unity or 100% and can thus be expressed as a percentage. A bootstrapping procedure using subsampling is implemented to generate confidence intervals for the RI estimates (Bickel et al., 2012). Details of the RI implementation are available in Roychoudhury et al. (2022).

## 2.7 Shapley Additive exPlanations using eXtreme Gradient Boosting (SHAP-XGBoost)

We use a robust ML technique called eXtreme Gradient Boosting (XGBoost) that gradually approximates and aggregates predictor-target relationships using subsets of a dataset (Chen and Guestrin, 2016). As in the MLR model, we train XGBoost on the predictors and target for each month (May-July), each GR, and each construct. XGBoost consists of multiple hyper-parameters that determine its performance. We use a Bayesian optimization technique called adaptive Tree-Parzen estimators (ATPE) to find the optimal hyperparameters by minimizing the squared error between the true SCF and predicted SCF (Bergstra et al., 2015). The model is trained until we achieve an $R^2$ (total explained variance) of 95% or more. In contrast to the MLR model (see Sect. 2.6), we do not explicitly define second-order terms for the predictors in the XGBoost model. Instead, we exploit the complex architecture of XGBoost to capture higher-order terms from the 22 main predictors to prevent user-defined bias during the training.

The ability of ML algorithms to model non-linear relationships between the target and predictors comes with the cost of decreased interpretability, given the intricate structure of XGBoost needed to model complex target-predictor relationships. The traditional model-dependent approach to interpret interactions in XGBoost models is through estimating feature importance and understanding decision pathways within the models (Jiang et al., 2009). Although various model-agnostic interpretability frameworks exist for complex ML models, we interpret our trained XGBoost models using the Shapley Additive explanation (SHAP) framework based on game theory, which quantifies the contribution of predictors and their interactions to the target response (Lundberg et al., 2020). Keeping in line with our definition of importance, we use SHAP to quantify the change in the target due to each predictor (the main effects) and their pairwise interactions. Thus, we can decompose the difference in predicted SCF into 253 ($N + {}^{N}C_2 = 22 + 231 = 253$ where $N = 22$) individual contributions for each XGBoost model, out of which 22 represent the main predictors and 231 represent the pairwise interaction contribution to the target. Here, pairwise refers to the contribution of one predictor in the presence of a second predictor. Instead of $R^2$ as in MLR, each SHAP value

(for a predictor) represents a fraction of the magnitude of SCF. The SHAP values were normalized to percentages, defined hereafter as SHAPc, by averaging the absolute SHAP values and dividing by their sum. This enables an analogous comparison to the RI metric as a percentage contribution to the total SCF (target) response. Additional details on this implementation are available in the Supplementary Information (Sect. S1.4).

## 2.8 Leveraging Model Constructs

We perform our analysis based on three model constructs: 1) the Observation-to-Model (Obs-Model) Construct, where SCF from MODIS is the target variable; 2) the Model-to-Model (Model-Model) Construct, where SCF from each reanalysis dataset is the target against corresponding predictors; and 3) the Observation-to-Observations (Obs-Obs) Construct, where we chose a set of variables directly observable through satellites (MODIS SCF, MODIS LST, MAIAC AOD, and IMERG PRECIP) to explore the non-linear sensitivities between SCF and its predictors depicted in Eq. (1).

The utility of these constructs lies in their ability to represent the true Earth system response based on the sensitivity of a target phenomenon, such as snowmelt, to its drivers (predictors) and their interactions. The Obs-Obs construct is considered the closest to ground truth, as it relies solely on observational data, but may be overestimated since not all potential drivers are currently observed. The Obs-Model construct can also depict the true response of these interactions, but has an inherent bias in the drivers that the models simulate. Conversely, the Model-Model construct captures the sensitivity of the target phenomenon to its predictors as defined by the model design, offering insights into the schemes and parameterizations defined in the model.

The regression approach from Eq. (1) is reserved only for the Obs-Model and Model-Model constructs, while the Obs-Obs construct is solely used to elucidate AMI that can be observed through satellites (see Sect. 3.4). This is due to the lack of a diverse range of predictors available from observations with consistent spatio-temporal coverage, a gap that can be bridged by reanalysis products.

Importance estimates in the Obs-Model construct encompass all three terms in Eq. (1) (especially the unresolved stochastic processes driving SCF in Term 3 of Eq. (1) and are the closest approximation to an observable estimate of AMI. The Model-Model construct sheds light on the representation of cryospheric processes driven by AMI in each reanalysis. Comparing the Obs-Model and Model-Model constructs can thus provide insights into the processes in Term 3 of Eq. (1) and highlight any potential misattribution of importance, or underrepresented processes estimated in any of the terms within the Model-Model construct.

## 3 Results

The motivation behind this analysis lies in the difference in SCF across satellites and reanalyses. In Fig. S1, we show the average spatial distribution of SCF in the late snowmelt season across four data sources (MODIS from satellite, ERA5-Land, MERRA-2, and MATCHA as reanalyses). SCF is very high in ERA-Land, which can be attributed to precipitation bias leading to excessive snowfall in the ECMWF snow model, while extremely low SCF in MERRA-2 can be attributed to the high snow depth specified in its land model to consider 100% SCF, leading to lower SCF (Orsolini et al., 2019). This disparity in SCF

across different datasets alludes to diverse model representations of processes driving SCF, which we try to leverage in this study. SCF in MATCHA resembles that of MODIS, which is a possible result of CLM-SNICAR coupling within MATCHA's model framework, effectively constraining SCF.

In this section, we quantify the importance of AMI on snow and analyze the variables driving these interactions. First, we demonstrate significant variability in SCF representation across reanalysis products (Sect. 3.1), then quantify AMI's contribution to SCF variability using two importance metrics (RI and SHAPc) that capture different interaction orders (Sect. 3.2). We identify key variables within AMI on snow by decomposing contributions from specific aerosol and meteorological subgroups (Sect. 3.3), then employ network analysis to visualize emergent connections within AMI, revealing significant differences between observational and model representations (Sect. 3.4). We discuss varying orders of interactions, coupling strength across reanalyses, and potential misattributions within interacting variables. Finally, we examine observable AMI (Obs-Obs construct) on snow using satellite observations to assess ground truth relationships (Sect. 3.5), revealing relationships between SCF and both land surface temperature, precipitation, and aerosol optical depth.

## 3.1 Quantifying Importance of AMI to SCF

As mentioned in Sect. 2, we consider 22 predictors spanning aerosols (AER), meteorology (MET), and elevation (ELEV) from the three reanalyses and regress them on SCF using statistical and machine learning regression methods. We focus specifically on the interacting terms between aerosols and meteorology, defined as aerosol-meteorology interactions on snow (AMI on snow). Our analysis is based on the importance estimates from the regression algorithms, which denote the sensitivity of the 22 predictors and their higher-order (second-order and/or more) terms to the target variable (SCF). This sensitivity is quantified by two metrics, relative importance (RI) from multi-linear regression (Sect. 2.6) and Shapley contribution (SHAPc) from ML (Sect. 2.7). We also use two model constructs on this regression framework to distinguish between the importance of AMI on snow from an observational (Obs-Model construct) and reanalysis (Model-Model construct) point of view. The key point to note is that the target variable (SCF) in the regression is used from two sources: 1) satellite data from MODIS for the Obs-Model construct, and 2) each reanalysis model for the Model-Model construct.

In Fig. 2, we show the RI and SHAPc importance distributions of AMI and MMI on snow in the Obs-Model (2b) and Model-Model construct (2c) for LSC regions in the late snowmelt season. The statistics (mean and standard deviation) of the importances for AMI on snow are summarized in Table 1. RI and SHAPc importances for MMI on snow are higher than those for AMI across all datasets, with an average contribution of 50-70% (both RI and SHAPc) to SCF variability. AMI on snow shows a consistent magnitude across all datasets in the Obs-Model construct with an average RI of 10-20% and SHAPc of 20-35%, indicating a significant contribution to SCF variability. In the Model-Model construct, the mean of the RI and SHAPc distributions for AMI on snow are lower by an average of 10% than in the Obs-Model construct. The spread in the importance distribution of AMI on snow across the constructs and datasets ($\sigma$ from 1.7 to 7.6 in Table 1) is higher than the difference in the mean importance of AMI on snow for both RI (difference $\mu$ difference by 4.5) and SHAPc (difference in $\mu$ by 8.2). A non-parametric Mann-Whitney test of the AMI on snow distributions (both RI and SHAPc) shows a significant difference (95% level) for both constructs across the three datasets. AMI on snow is thus significant for both constructs, and a lower AMI

on snow importance in the Model-Model (relative to Obs-Model) construct suggests second and/or higher-order interaction terms that may be missing or unresolved within the reanalysis model framework (see Sect. 2.5). The large spatio-temporal variability of SCF (Fig. S1) in the late snowmelt season, combined with the difference in AMI's importance to snow across both constructs, suggests the disparity in AMI-related processes that drive SCF within each reanalysis dataset. SHAPc values for AMI on snow are higher in both constructs compared to RI, which can be due to the ability of XGBoost to capture the non-linear interactions to a fuller extent, compared to the MLR model, where the interactions are restrictive in their definition (only non-square product terms).

## 3.2  Key Variables within AMI on Snow

We further decompose the importance of AMI on snow in both constructs by meteorology (MET variables with five subgroups of variables) and aerosols (AER variables with four subgroups of variables) in Fig. 2a and 2d. Among AER variables, carbonaceous aerosols and total AOD at 550 nm (Others) contribute significantly to the AMI on snow importance (average 18% and 14% respectively), followed by dust (average 11%) in both the constructs and metrics. Among MET variables, circulation-related variables contribute the highest (average 13%), followed by cloud cover variables (average 10%).

An alternate way to visualize the contribution of each subgroup of variables across AER and MET predictors is shown in Fig. 3. This contribution (also expressed in %) is the importance values ($\alpha$ in %) of the AER and MET predictors normalized to the total mean importance of AMI on snow as mentioned before. We see that circulation variables contribute the most (38%) to AMI on snow in the Obs-Model construct, whereas radiation and temperature dominate (23%) in the Model-Model construct. Carbonaceous variables are dominant across both constructs (30%); however, dust contributes more in the Obs-Model construct (24%) than in the Model-Model construct (20%). Additionally, the AER subgroup Others (including total AOD at 550 nm and surface sea-salt) makes a significant contribution, primarily driven by total AOD at 550 nm.

The prevalence of carbonaceous aerosols can be attributed to increased surface BC and total aerosol optical depth (AOD) in the vicinity of the LSC regions during the pre-monsoon season (April-May). This includes wheat crop residue burning in the northern part of the Indian subcontinent, inducing potential interactions with large-scale synoptic atmospheric circulation in the subsequent months (late snowmelt season) that lead to changes in near-surface temperatures, convection, and accelerated melting (Das et al., 2022; Kumar et al., 2011; Lau et al., 2006; Ramanathan et al., 2007; Lau and Kim, 2018). Such interactions can also allude to the deposition of LAPs through interactions between aerosols, geopotential height, and near-surface variables. Multi-model intercomparison of global aerosols has also reported that carbonaceous aerosols contribute an average of 70% to aerosol-induced absorption, a key process in AMI on snow (Sand et al., 2021). A higher importance can be seen across the predictor subgroups and datasets in AMI on snow distributions within the Obs-Model relative to the Model-Model construct for both metrics. This can be attributed to the absence of unresolved processes and their interactions driving SCF in the model representation of the three reanalyses. It can, however, be the case that the observations are biased or that the modeled SCF might not be spatially or temporally in phase with MODIS SCF. With the current observing system, we cannot attribute this difference in importance to the errors in the observations, the models, or a combination of them.

The metrics, RI, and SHAPc highlight two aspects of these interactions. The SHAPc distribution of AMI's importance onto snow has a higher spread ($\sigma$ between 5.6 - 7.6), indicative of a bulk non-linear effect. This is seen in Eq. (1) where SHAPc reflects the sensitivities in all three terms. Whereas for RI, the lower spread in the distribution of AMI's importance on snow ($\sigma$ between 1.7 to 3.2) indicates specific (local) second-order processes captured by RI (first and second term in Eq. (1)).

## 3.3 Emergent Connections within AMI

Moving beyond the groups of AER and MET predictors that dominate AMI on snow, we visualize the individual interactions within AMI on snow (Fig. 4) using concepts from network analysis (Inglis et al., 2022). In particular, we visualize the average of the RI and SHAPc importances to emphasize the overall importance of each interaction to SCF. We do this across both model constructs, both importance metrics, and three reanalysis datasets (thus 12 networks). For each of the 12 networks, the six larger nodes (circles) represent the AER predictors while the smaller nodes represent the MET predictors. The connections (edges) between the nodes are weighted by the pairwise importance according to the importance metric (RI and SHAPc) to represent the interactions (edge connections) and their strengths (edge widths and colors) between AER and MET variables on the snow interface. The node sizes depend on the degree of each node (number of edge connections per node, weighted by the edges). For a total of 21 predictors, these would lead to $\frac{^{21}C_2}{2} = 105$ edges across 21 nodes for each network. These edges are weighted by their color and width according to their interaction importance (between 1 to 100%). For the network analysis, we have used the `networkx` package (v2.8.4) and Gephi (v0.10) for the network graph layouts.

Since we are considering interactions within AMI on snow, we consider the pairwise interactions of six AER predictors with 15 MET predictors. As such, the degree of the AER nodes will always be much higher than that of the MET nodes. The concept of weighted degree is defined in Appendix A. This can be seen from the networks in Fig. 4, where the AER nodes have a larger size relative to the MET nodes. In the following sections, we primarily focus on the weighted edges of these networks as the degrees of the AER nodes are relatively similar across the networks. It is important to note here that the network edges that represent the strength of the interaction are based on the importance metrics (RI and SHAPc) that are calculated based on the 21 aerosol and meteorological predictors. As such, considering a different set of predictors might thus influence these importance values and hence the network edges, which can obscure certain interactions that are not captured due to the choice of the predictors. We further explore the possible idea of misattribution of these interactions in Sect. 3.3.4.

### 3.3.1 *Observed* versus *Model* Snow Interface

A prominent feature across all the networks is the difference between the interactions seen between the two constructs across the three reanalyses. The networks in the Obs-Model Construct show a higher number of *strong* (> 50% importance, moderate to very high) interactions, whereas the networks in the Model-Model construct show fewer and specific *strong* interactions. This suggests greater interaction strength/importance between the AER and MET predictors that contribute to the target (SCF) variability compared to what the models in each reanalysis show. The higher density of connections within the Obs-Model construct suggests significant AMI-related interactions at the *real snow interface*, compared to what the models in the reanalyses consider to be relevant for the *model snow interface*. This agrees with our observations from Fig. 2 and Table 1, where the

distribution of AMI's importance on snow in the Obs-Model construct is statistically significant compared to that in the Model-Model construct. Additionally, as discussed in Sect. 2.8, the interactions shown in the Obs-Model construct can reflect physical reality, while the Model-Model construct only captures the interactions that the model frameworks parameterize within themselves. In Table 2, we present the dissortativity of the networks depicted in Fig. 4 (Newman, 2002) (defined in Appendix A). Dissortativity measures the heterogeneity in connection patterns between variables (nodes) of different importances. It indicates whether highly connected nodes prefer to interact primarily with less connected nodes (high dissortativity) or with other highly connected nodes (low dissortativity). Variations in dissortativity, therefore, reflect how well the networks capture the hierarchical structure of AMI on snow. Across nearly all reanalyses (Table 2), Obs-Model networks exhibit consistently more dissortativity (–0.5 to –0.9) than Model-Model networks (–0.4 to –0.7), with the largest difference (0.3) in ERA5/CAMS4 and MERRA-2. Higher dissortativity indicates that observed AMI on snow shows more connections between AER and MET variables of different importances than those within the reanalyses. This suggests that models within the reanalyses underrepresent the complexity of AMI on snow, oversimplifying the connections between key AER and MET predictors. This confirms that real-world AMI on snow is far more complex than what current models can fully capture, potentially contributing to biases in modeled snow processes.

It is important to note here that the dissortativity of the networks also indirectly depends on the skill of the predictors, and their interactions in capturing accurate sensitivity to the target variable, which impacts the importance metrics (RI and SHAPc) and thus the edge of the networks (using which dissortativity is calculated). This skill reflects the accuracy of the predictors in representing correct atmospheric conditions and their inherent noise and internal variability. Noise in reanalysis products, which serve as our observational proxies, can potentially manifest as additional/stronger/weaker connections in the networks, thus affecting dissortativity. However, the consistency of higher dissortativity values across different reanalyses and different importance metrics (from Table 2) suggests that the observed differences primarily reflect real structural limitations in how models capture the hierarchical complexity of aerosol-meteorology interactions on snow, rather than just artifacts of data noise. Furthermore, the dissortativity differences (up to 0.3) between Obs-Model and Model-Model constructs indicate that these differences represent meaningful structural variations in aerosol-meteorology interactions that exceed what would be expected from noise alone.

To highlight the difference between the two constructs, Fig. 5 shows the aggregated (summed) interactions between AER and MET variables for each construct and importance metric across all three reanalyses. This demonstrates the interactions that each construct generally emphasizes. We also show the positive difference in the interactions between the Obs-Model and the Model-Model construct (in Fig. 5c) for each metric, which can highlight specific interactions missing in the modeled reality (Model-Model construct). Both RI and SHAPc emphasize interactions of surface dust (DU) with circulation variables (particularly geopotential height at 300 hPa and 500 hPa, as well as mean sea level pressure) in the Obs-Model construct, which are weaker in the Model-Model construct. We see this in Fig. 5c, where both difference networks for both metrics highlight strong interactions with the circulation variables, suggesting that interactions of circulation variables, particularly with dust, are missing in the Model-Model construct. RI also shows missing interactions with temperature variables in the difference

network, which is not visible for SHAPc. On the other hand, SHAPc emphasizes moderate-high (50% to 75%) interactions with cloud cover variables (particularly medium, high, and total cloud cover) that are missing in the Model-Model construct.

### 3.3.2 Varying Orders of Interactions

Comparing the networks between the RI and SHAPc metrics provides insights into how each of these two metrics highlights the functional aspect of AMI on snow. As mentioned in Sect. 2.5, RI, and SHAPc highlight different orders of interactions based on their definitions. In Fig. 4, we can see that the RI importance metric focuses more on specific interactions, while the networks for the SHAPc metric appear more interconnected, with a broader distribution of importance values. In Fig. 5, we see that interactions of surface dust (DU) are the strongest across both metrics, but RI emphasizes product interactions with temperature and surface energy variables (particularly surface sensible heat flux or sShf), while SHAPc captures the higher-order interactions with temperature across both constructs. Both metrics fail to capture interactions with circulation, which can be seen in the difference networks in Fig. 5c. This emphasizes the need to include circulation-related interactions in the model frameworks of all three reanalyses. From the difference networks, we can visualize how SHAPc and RI metrics differ between the constructs and highlight higher-order processes that are inadequately represented in the model framework of each reanalysis. We see that *strong* second-order interactions (from RI) of DU and AOD550 with temperature and circulation variables are underrepresented in all three reanalyses, while *strong* higher-order interactions of AER variables with cloud cover and circulation are missing across all the reanalyses. However, SHAPc does capture temperature interactions in both constructs. Higher dissortativity in the three reanalyses (Table 2) for the SHAPc metrics in the Obs-Model construct suggests a greater variety of higher-order processes across AER and MET predictors at the *observed* snow interface compared to the *model* snow interface.

### 3.3.3 Coupling Strength across the Reanalyses

From Fig. S1, SCF from MATCHA agrees most with the observed SCF during the study period (May to July). This can be attributed to the stronger coupling within MATCHA's model framework, which couples aerosols, radiation, and snow, in comparison with the other two reanalyses. This is also reflected in the density of the connections observed in MATCHA, particularly in the Obs-Model construct from Fig. 4a, relative to that of ERA5/CAMS4 and MERRA2. Despite the tighter coupling in MATCHA, there are notable differences between the networks across the constructs. In the individual networks in Fig. 4, the Model-Model construct for MATCHA emphasizes interactions with carbonaceous aerosols, whereas the Obs-Model construct highlights dust (DU). Ideally, the interactions in the Model-Model construct should be similar to those in the Obs-Model construct. However, significant differences between the constructs across all the reanalyses can indicate interactions that are inadequately represented in each reanalysis. We explore these differences further in Fig. 6, where we show the underrepresented interactions across all the reanalyses (using the positive differences between the importance seen in the Obs-Model and the Model-Model construct aggregated across RI and SHAPc). For MATCHA, the major deficiency lies in representing interactions of DU and carbonaceous aerosols with circulation variables (particularly geopotential height and mean sea level pressure). Further analysis of these interactions (as shown in Fig. S2) reveals that MATCHA fails to adequately represent

the second-order interactions (based on RI) of DU with circulation, temperature, and surface energy variables, as well as the higher-order processes (based on SHAPc) involving absorbing aerosols with circulation and cloud cover variables. We show the underrepresented interactions (positive difference between Obs-Model and Model-Model construct) for different orders across each reanalysis, based on RI and SHAPc in Fig. S2.

While MATCHA exhibits denser connections in the Model-Model construct compared to ERA5 (Fig. 4), the density of the interactions for MERRA2 is closely comparable to that of MATCHA, despite differences in individual interactions. This is further evident in Fig. 6, where the underrepresented interactions in MATCHA and MERRA-2 are significantly less compared to ERA5/CAMS4, indicating stronger coupling in both models. Both MERRA-2 and ERA5/CAMS4 show insufficient interactions with circulation and temperature variables. However, ERA5/CAMS4 exhibits a greater deficiency in these interactions,

extending to cloud cover as well. Detailed analysis of the missing interactions based on their order (across RI and SHAPc from Fig. S2) reveals that in MERRA-2, second-order interactions between DU and AOD with circulation and temperature variables are absent, as well as higher-order processes between dust and geopotential height. In ERA5/CAMS4, the density of the underrepresented interactions in Fig. 5 shows that a large number of interactions are not represented adequately in its model framework, with significant gaps in second and higher-order processes involving cloud cover. Dissortativity values from Table

2 show that while all three reanalyses show higher diversity of AMI on snow in the Obs-Model construct, both MERRA-2 and MATCHA have the highest dissortativity in the Obs-Model construct, especially for higher-order processes (represented BY SHAPc).

    Another approach to understanding the inadequate processes in the reanalyses is to compare the interactions of ERA5/CAMS4 and MERRA-2 across the two constructs with those of MATCHA. Given the strongly coupled nature of MATCHA, Fig. S3

specifically highlights predictor interactions that ERA5/CAMS4 and MERRA-2 fail to capture compared to MATCHA. The interactions are estimated as before by taking the positive difference between the importance of ERA5/CAMS4 and MERRA with MATCHA for each construct. Both constructs demonstrate a lack of interactions with circulation variables in the two reanalyses, shown by the presence (absence) of strong edge connections between circulation and AER variables in the Obs-Model (Model-Model construct). Additionally, meteorological interactions with DU are more pronounced in the Obs-Model

construct, in contrast to their almost minimal contribution in the Model-Model construct, where carbonaceous aerosols are more significant. The networks indicate that significant interactions of aerosols with circulation variables should be present in both reanalyses. However, MATCHA also fails to adequately capture the circulation interactions as seen in Fig. 6. Specifically, ERA5/CAMS4 should focus more on circulation interactions with DU, while MERRA-2 should emphasize interactions with carbonaceous aerosols to capture the coupling within MATCHA.

While these aggregated networks (in Fig. 6) can highlight the higher-order interactions within each reanalysis, it is necessary to consider the skill of these datasets in accurately depicting these real-world interactions, which can arise from the absence of parameterizations in their respective model frameworks to represent these interactions, or misrepresenting the order of interactions between these predictors (2nd order or higher) if at all present within the reanalysis framework.

### 3.3.4 Potential Misattributions in each Reanalysis

In addition to highlighting the underrepresented interactions through the difference networks, we hint towards potential misattributions in each reanalysis through Fig. 6 by examining interactions that are strong in the Model-Model construct but absent in the Obs-Model construct. This is estimated using the negative difference of importances between the Obs-Model and the Model-Model construct (shown through red edges) instead of the positive difference for the underrepresented interactions (through black edges). These discrepancies highlight significant interactions and processes that the models consider impor-

tant for the *model* snow surface but cannot capture for the *observed* snow surface. Specifically, we find that MERRA-2 and MATCHA overemphasize the interactions between dust (DU) and accumulated precipitation (PRECIP), while ERA5/CAMS4 places undue importance on the interactions between dust (DU) and skin temperature (SKT), even though interactions with 2-m temperature is much more significant in the Obs-Model construct. Although these interactions are related to temperature, the disparity here suggests that feedbacks between surface dust aerosols and near-surface temperature are more significant than

those involving the surface itself.

An associated issue with misattribution is the buffering of the snowmelt response from one predictor due to the presence of other predictors, which can obscure the true impact, especially of the aerosol predictors on snowmelt, resulting in inaccurate conclusions about their relative contributions. Buffering of the interaction sensitivity by other dominant predictors is seen in aerosol-cloud-precipitation interactions, where different cloud processes can buffer the sensitivity of aerosols to precipitation

(Stevens and Feingold, 2009; Michibata et al., 2020). Although we can potentially highlight where each model misattributes the snowmelt sensitivity for AMI interaction, we are unable to determine the buffering of the snowmelt response of AER predictors by the MET variables with the current approach. As mentioned previously, the interpretation of misattribution of these interactions within each reanalysis depends on the accuracy of these predictors in representing their sensitivity to SCF, as well as the representation of these feedbacks within the reanalysis frameworks.

### 3.3.5 Bringing it altogether

The networks reflect the complexity within each reanalysis that reflects the feedbacks between the AER and MET variables for both observable and modeled realities (the constructs). The progression in importance, in terms of both the number and strength of interactions from ERA5/CAMS4 to MATCHA (both number and strength of interactions) from ERA5/CAMS4 to MERRA-2 TO MATCHA across both constructs signifies the degree of coupling incorporated in the three reanalyses, This

progression reflects the absence of coupling between aerosols and meteorology in ERA5/CAMS4, in contrast to MERRA-2 and MATCHA.

The degree, or the number of relatively stronger connections to a node (predictor), reflects the strength of the coupling processes in ESMs, both direct and indirect. However, the edge strength is a function of the abundance (magnitude) and co-variability of the interacting predictors (nodes) and indicates the importance of the coupling/interaction. Visualizing the

number and strength of each interaction within AMI on snow through the network diagrams highlights relevant processes of different orders driving SCF during the study period. These interactions are otherwise difficult to disentangle due to their

inherent complexity. Using constructs and different metrics of importance allows us to demonstrate which interactions and their complexities are necessary to be represented in each reanalysis model. Additionally, comparing these networks helps identify the misattribution of interacting processes in each reanalysis. This can be analyzed from the interactions present in the Model-Model construct but absent in the Obs-Model construct. We see that interactions between DU and PRECIP in MERRA-2 and MATCHA are given unnecessary importance, while for ERA5/CAMS4, it is the interactions between DU and SKT that are overemphasized. Previous studies have identified significant biases in model representations of dust processes, such as overestimation of precipitation's impact on dust abundance as well as large variability in dust simulations across models (Pu and Ginoux, 2018; Kok et al., 2017; Zhao et al., 2022), and the complex interactions between dust aerosols and surface temperature that can lead to biased parameterizations of near-surface processes (Stante et al., 2023). Overall, the need to incorporate large-scale circulation-related interactions is emphasized across the reanalyses to correctly simulate SCF during the study period. Uncertainties associated with atmospheric circulation are a pertinent problem across climate models due to internal variability of the Earth's climate and errors in model representation (Shepherd, 2014). Interactions of the large-scale circulation dynamics with unresolved small-scale processes involving clouds, convection, boundary layer, complex topography, and near-surface temperature remain uncertain across models (Stevens and Bony, 2013; Bony et al., 2015; Holtslag et al., 2013; Sandu et al., 2019). Considering aerosols adds to this uncertainty due to interactions with cloud microphysics, precipitation, and convection, making accurate representation even more challenging (Bony et al., 2015; Dagan et al., 2023; Fan et al., 2012; Mülmenstädt and Wilcox, 2021). Anthropogenic forcings due to greenhouse gases and aerosols cannot be neglected as they have been shown to influence trends in circulation variables like geopotential height at 500 hPa and mean sea level pressure (Christidis and Stott, 2015; Gillett et al., 2013; Ming and Ramaswamy, 2011). Improving the circulation-related interactions with aerosols in coupled ESMs can improve the representation of monsoon, regional, and local aerosol transport pathways, aerosol deposition, and cloud distribution in complex regions like HMA (Li et al., 2016; Hu et al., 2024; Mülmenstädt and Wilcox, 2021).

The underrepresented dust-circulation interactions from our network analysis align directly with the physical mechanisms described by (Lau and Kim, 2018) regarding the snow-monsoon relationship in Asia. Their modeling experiments showed that the deposition of dust on snow initiates a series of interconnected processes: reduced snow albedo, increased solar radiation absorption, and accelerated snowmelt, with subsequent modification of regional circulation patterns. Specifically, they showed that dust deposition in April-June leads to atmospheric warming and pressure patterns (changes in geopotential height) that enhance dust transport to the Himalayan-Indo-Gangetic region. Our network diagrams reveal that these critical connections between dust and circulation variables (particularly geopotential height and mean sea level pressure) are insufficiently captured across all three reanalyses, despite being important for SCF variability. This explains persistent biases in SCF and dust, particularly in LSC regions nearest to major dust sources like the Taklamakan Desert (see Fig. S1 and S4) (Zhao et al., 2022), which Lau and Kim identified as contributing significantly to dust deposition on Himalayan snow. (Zhao et al., 2024) confirms the role of dust in impacting the Asian summer monsoon and how more accurate dust simulations can help constrain the monsoon circulation patterns. The progression in interaction complexity we observe from ERA5/CAMS4 to MATCHA shows

improvement, but still indicates insufficiencies in representing the dust-snow-circulation feedbacks that are crucial for regional climate dynamics.

Both BC and dust impact snow by modifying the snow albedo feedback, although their relative importance to radiative forcing remains uncertain, as mentioned in Sect 1. We see in Sect. 3.2 that while BC (as a component of carbonaceous aerosols) dominates the bulk contribution to AMI's importance on SCF, individual interactions of meteorology variables with DU become more prevalent when we analyze the networks after decomposing this bulk contribution to AMI. This is also seen across the networks, where the node sizes for BC and DU (based on their weighted degree) are similar, reflecting a similar number of interactions with the meteorology variables. We allude to this disparity in Fig. S4, where we see a higher abundance of DU (mostly natural sources) than BC (mostly anthropogenic sources) over HMA. The spatial distribution shows that higher values of surface BC are primarily concentrated in the vicinity of the glacier regions, indicating pollution sources from nearby Asian countries. A higher concentration of surface DU is concentrated in northern HMA, especially in the LSC regions, due to its proximity to the Gobi and Taklamakan deserts. The monthly variations of BC and DU also show a greater abundance of DU compared to BC during the study period. The prevalence of inadequate representation of dust and circulation-related interactions can thus indicate biases in the model to simulate the abundance of these quantities over the LSC regions in HMA, compared to the biases in simulating BC abundance.

### 3.4   Observable AMI on Snow

Given the significance of AMI in regulating SCF during the study period, it would be useful to interpret what these interactions within AMI on snow represent through observed relationships between the predictors and SCF. While having multiple predictors from satellite observations would be ideal for exploring AMI on snow (or any Earth system interactions) to its fullest extent, we consider four such variables from satellite observations, MAIAC AOD, MODIS LST, and IMERG PRECIP, and visualize their relationship with MODIS SCF, which we defined as the Obs-Obs construct in Sect. 2.8. Exploring the relationship between predictors in the Obs-Obs construct will provide a basis of ground truth for relative comparison with findings in the other two constructs, and aid in understanding the relationships between the chosen predictors and their SCF response.

In Fig. 7, we show the relationship between MODIS SCF and the predictors MODIS LST and IMERG PRECIP, weighted by the distribution of MAIAC AOD for LSC regions during May-July. We observe an overall trend of exponential decay of MODIS SCF with MODIS LST above 0°C, dominated by high values of MAIAC AOD, especially at higher LST and lower SCF. Such behavior can point to the radiative effects of absorbing aerosols, causing warmer temperatures (high LST) and accelerated snowmelt (low SCF). However, this does not imply causality as it might be attributed to the warmest areas in the domain (with high LST) located at lower elevations and directly affected by air pollution (high AOD), or that spatial resolution of 0.75° in the datasets might include non-snow-covered regions with high temperatures. We use a mutual information-based metric to quantify the bulk non-linear association of SCF to LST (as shown by the bars in Fig. 7) (Kraskov et al., 2004). The strongest association between MODIS LST and SCF occurs for low to moderate values of AOD (values within $0.04 - 0.10$). Compared to satellite observations, MATCHA shows a similar relationship between SCF and LST compared to the other two datasets (for both Obs-Model and Model-Model constructs). Given that MATCHA is the only framework among the three datasets

with coupling between snow, radiation, and LAPs, this similarity confirms the ability of parameterizations in MATCHA to represent AMI on snow better than datasets from MERRA-2 or ERA5/CAMS4. In the Model-Model construct, MERRA-2 shows the strongest relationship between SCF and LST at moderate to very high values of AOD (0.1 – 6.8), suggesting an overestimated aerosol loading (compared to AOD in Obs-Obs) in LSC regions during the late snowmelt season that might contribute to the lower SCF values over HMA seen in Fig. S1 for MERRA-2. On the other hand, ERA5/CAMS4 has stronger

SCF-LST sensitivities for values below high AOD (< 0.21), which can allude to a lack of coupling related to aerosol radiative feedbacks within the ERA5 model that translates to an absence of strong SCF-LST dependencies to the AOD distribution in CAMS-EAC4. The strong SCF response to LST for ERA5 at lower AOD values (below high AOD, <0.21) can either allude to the buffering of aerosol effect reflected in the lack of coupling within ERA5 related to aerosol radiative feedbacks. This can thus indicate misattribution of the SCF response to aerosols in the presence of meteorology.

We also see an exponential decay between MODIS SCF and IMERG PRECIP in the Obs-Obs construct, with strong SCF-PRECIP dependency at low to moderate values of AOD (0.04 – 0.10). This might indicate potential removal (wet scavenging) of absorbing aerosols by precipitation that can result in lesser amounts of exposed absorbing aerosols onto snow, hence reducing snow darkening and its impact on snowmelt (Gryspeerdt et al., 2015). As mentioned earlier regarding the SCF-LST relationship, direct causality is not implied as the wetter areas in the domain (high PRECIP) can be located at lower elevations

and directly impacted by air pollution (high AOD), or that the spatial resolution of 0.75° might include areas with non-snow-covered regions with high precipitation. MATCHA also exhibits the most similarity in the SCF-PRECIP relationship to the Obs-Obs construct compared to the other two datasets, with strong sensitivity of SCF to PRECIP in the low-moderate AOD range. This reflects the degree of coupling within MATCHA compared to MERRA-2 and ERA5/CAMS4. MERRA-2 reflects the underestimation of SCF as seen in Fig. S1, and the SCF-PRECIP relationship is strong for moderate to very high values

of AOD (0.1 – 6.8) compared to the other datasets (where the SCF-PRECIP is strong for values below high AOD), suggesting overestimated AOD within the model (compared to AOD in Obs-Obs) in LSC regions during the study period. High SCF in ERA5/CAMS4 is dominated by low to moderate AOD (0.04-0.10) in the Model-Model construct (as compared to high SCF when AOD is low or < 0.04 for Obs-Obs). This can indicate higher-than-usual aerosol loading within CAMS-EAC4 in the study region during the late snowmelt season.

## 4  Summary and Implications

### 4.1  Main findings

We evaluated three state-of-the-art reanalysis frameworks in their ability to capture a particular case of Earth-system interactions, those pertaining to feedbacks between aerosols and meteorology that affect snowmelt over HMA. By employing a data-driven approach across twenty-two distinct geophysical quantities (six aerosol and 15 meteorological), we assessed

aerosol-meteorology interactions at the snow interface (AMI) over low snow-covered regions (LSC) of HMA through interactions of various orders of these variables. Our main findings are as follows,

1. **Importance of AMI to SCF Variability.** We estimated the importance of AMI on snow in driving SCF variability across three reanalyses and two importance metrics during the late snowmelt season, building on our previous work (Roychoudhury et al., 2022). While interactions within meteorology at the snow interface (MMI) contribute the most to the variability of SCF (∼60% contribution), drivers related to AMI account for an average of 20% of the SCF variability. The robustness of the importance of AMI on snow was established by using 1) two regression-based algorithms: one statistical and one machine learning-based, 2) using model constructs to distinguish between model versus observable relationships that impact SCF, and 3) using three state-of-the-art reanalysis with varying degrees of coupling parameterizations, to quantify the importance of non-linear interactions to snowmelt and characterize them within AMI.

2. **Significant Drivers within AMI on snow.** By introducing the concept of constructs for the regression algorithms that correspond to observed and model reality (within each reanalysis), we determined which group of aerosols and meteorology variables contributes most to the importance of AMI on snow. Dominant contributions from carbonaceous aerosols (30%), dust (24%), and large-scale circulation variables (38%) contribute to AMI at the *observed* snow interface, whereas variables related to near-surface temperature (22 %) and surface energy fluxes (23%) are given priority at the *model* snow interface.

3. **Underestimation of AMI on snow across the reanalyses.** Comparative analysis between the constructs through network visualizations reveals 1) individual interactions between aerosols and meteorology variables that are underrepresented in each reanalysis and 2) the underestimation of AMI's importance to SCF within the reanalyses compared to satellite-based SCF, which highlights a significant disparity between observed and modeled data. Furthermore, by applying the concept of assortative mixing in networks (Newman, 2002), we can observe differences in the diversity of AMI interactions across both constructs for each reanalysis.

4. **Underrepresented interactions within AMI on snow.** Circulation-related interactions with dust aerosols, particularly those involving geopotential height and mean sea level pressure, are found to be significant yet insufficiently represented in the models within each reanalysis. Previous studies have mentioned the uncertainty with dust and circulation across models and how the interactions between the two initiate feedbacks affecting monsoon and snowmelt in HMA (discussed in Sect 3.3.5). The importance of circulation-related interactions suggests that interactions of absorbing aerosols and smaller sub-grid processes with large-scale atmospheric circulation involving clouds, convection, and transport across the boundary layer need to be addressed for more accurate snow hydrology and understanding of Asian monsoon dynamics.

5. **Complexity of coupling across each reanalysis.** Results suggest that reanalysis from NSF NCAR (MATCHA) strongly resembles the relationships between aerosols and meteorology to observed SCF, considering that the degree of coupling parameterizations interfacing the atmosphere and land (cryosphere) is highest in MATCHA due to the inclusion of feedbacks in its model between aerosol, radiation, and snow through CLM-SNICAR. Using available aerosol and meteorology observations from satellites in Sect. 3.4 also shows that MATCHA captures the joint sensitivities between aerosol and meteorology variables observed across satellites. Although both MERRA-2 and MATCHA incorporate some

degree of coupling within their models, our study suggests that interactions of dust with circulation variables would need more attention within the two. The models in both these reanalyses seem to overemphasize interactions of aerosols (particularly dust) with daily accumulation precipitation, instead of coupling with circulation variables such as geopotential height and mean sea level pressure. From our interpretation of the networks, it seems that ERA5/CAMS4 relies extensively on its non-coupled model framework and assimilation of observations and needs extra attention to circulation and cloud cover-related interactions in the future development of the ECMWF model. The variability in the importance distribution of AMI on snow across the reanalyses is also lower than the difference in the variability of AMI's importance on snow from both constructs (Fig. 2 and Table 1), indicating that the coupling within MATCHA is far from ideal. Thus, the need for parameterizations that represent the feedbacks between snow and aerosol abundances, including relevant snowmelt drivers like circulation-related variables, is necessary to consider in the development of future ESMs.

6. **Physics-informed insights.** The consistent importance of aerosol-meteorology interactions on snow over HMA across two regression algorithms and constructs suggests that the sensitivities observed of these interactions to snowmelt are not merely statistically inferred, but rooted in physics-informed insights. Available observations from satellites confirm these insights by demonstrating similar relationships between aerosol and meteorology variables, especially the strongly coupled reanalysis (MATCHA) as seen in Sect. 3.4. The radiative effect of absorbing aerosols, as well as wet scavenging of these aerosols by accumulated precipitation, is seen across the observations and the reanalyses, in addition to their inherent biases in representing these processes.

## 4.2 Implications to Earth System Predictability

The synergistic approach combining 1) model constructs, 2) statistical regression and machine learning methods, and 3) network analysis serves as a viable complement to conventional, resource-intensive feedback separation methods used within the community. This highlights the potential of our methodology to detect non-linear relationships not only at the atmosphere-cryosphere interface but also across other Earth system processes. Beyond quantifying the relevance of these coupled processes, this approach allows us to identify key variables driving these interactions and pinpoint deficiencies in their representation across different models. While current benchmarking frameworks for evaluating ESMs utilize diverse statistical metrics and assess models' ability to represent different climate modes of variability, our approach can assist in identifying specific interactions that are highly uncertain and complex for any Earth system phenomena, extending beyond snowmelt in the Third Pole (Lee et al., 2024; Lauer et al., 2020).

Our results emphasize the need to 1) incorporate relevant non-linear interactions involving circulation, temperature, and cloud cover between aerosol and meteorological variables within ESMs for improved predictions of snow hydrology, 2) inform specific variables that need to be assimilated in the design of observing systems, and 3) include a broader array of observable variables across different Earth system components (e.g., aerosols and meteorology in this study) in future phases of the Coupled Model Intercomparison Project (CMIP) and its related ESM-SnowMIP initiative, to assess co-variability across both aerosols and meteorology (Krinner et al., 2018). Based on our findings in Figs. 2-3 and the individual interactions within the networks in Figs. 4-6, we emphasize the incorporation of variables related to large-scale atmospheric circulation, near-surface

temperature, as well as refined proxies for absorbing aerosols, particularly dust, in future CMIP and ESM outputs. Considering the future direction of ESMs toward Integrated Earth System Model and Analysis (IESM, IESA) with an emphasis on observationally constrained coupled chemistry-climate models (CCMs), joint assimilation of AMI-relevant variables is essential for this development (Bocquet et al., 2015; National Academies Press, 2018). Furthermore, with the advent of ML forecast models such as GraphCast trained on ERA5 data (Lam et al., 2023), it is more important than ever to assess how existing reanalyses represent the representation of coupling of relevant interactions. Diagnosing and quantifying the strength of such interactions across Earth system processes is essential to reduce uncertainties in Earth system predictions and to minimize the false attribution of observed environmental changes (National Academies of Sciences, Engineering, and Medicine, 2022; Ripple et al., 2023). Identifying underrepresented interactions in ESMs has the potential to enhance medium-range and sub-seasonal to seasonal forecasts of high-impact weather events, particularly water cycle extremes that can support greater resilience of vulnerable populations in climate-sensitive regions such as HMA (NOAA Science Advisory Board, 2021).

## 4.3 Limitations and Future Directions

It is important to recognize that while this study specifically analyzes the coupling among aerosols, meteorology, and snowmelt over HMA, our primary objective is to demonstrate a systems-science approach that can assist in unraveling interactions within any Earth system phenomenon, identify its key drivers, and highlight the inconsistencies among different models in their representation of coupled Earth system processes. This study serves as a proof-of-concept for employing a variety of methods (statistical, ML, model constructs, and network theory) to identify the biases in the representation of Earth system interactions within ESMs. While our analysis focuses on low snow-covered regions in HMA and the late snowmelt season, further assessments of AMI in high snow-covered (HSC) regions within HMA and during the snow accumulation period are also necessary. The more transient changes in seasonal snowpacks within LSC regions exhibit significant sensitivity of AMI to snow, whereas the non-seasonal snowpacks in HSC regions are influenced by longer timescales (Liu et al., 2021a). Preliminary results for HSC regions, shown in Fig. S5, also highlight the strong importance of AMI on snow in these regions, albeit with higher variability (reflected in the spread of the AMI distribution) compared to LSC regions. Additional observational datasets for SCF and other snow properties (e.g., snow albedo) also need to be explored to improve the robustness of our findings (Wu et al., 2021; Liu and Margulis, 2021; Rittger et al., 2021). Observational datasets across all the predictors, when possible, also need to be explored to strengthen our findings and facilitate a detailed analysis of the process-level physics underlying these interactions, as discussed in Sect. 3.4.

While our focus is on AMI on snow over HMA, a similar analysis of the interactions within meteorology and with elevation will provide further insights into these complex processes and their representation in models over the Third Pole. Beyond improving ESMs, our methodology can also inform a more optimal design of field campaigns in climate-vulnerable regions such as HMA. By identifying the key variables and interactions that affect snowmelt (or any other phenomenon of interest) and determining when and where these interactions are more pronounced, our approach can help optimize the allocation of limited observational resources. This would enable more strategic selection of the variables, locations, and periods to yield the most valuable information on critical Earth system processes in these regions. In addition, exploring the uncertainty within

the networks would be valuable, allowing us to quantify the confidence in the inferences drawn in this study. Observational datasets for predictors that are currently derived primarily from reanalyses are also needed to conduct a more comprehensive analysis, beyond what we presented in Sect. 3.4, where we evaluated the joint distributions of interacting variables in relation to the target variable.

We acknowledge that the insights from our approach are dependent on the choice of variables to represent these processes. Thus, future studies incorporating other relevant variables (e.g., net surface radiation fluxes, boundary layer height, vertical profiles of thermodynamic variables, and wind) will also be valuable. Moreover, reanalysis datasets are known to exhibit biases, particularly in high-elevation regions (see Sect. 2.1); hence, their relative skill and accuracy must be taken into account when interpreting their sensitivity to the target variable. An additional avenue to explore is the separation of misattribution and buffering among the drivers in the identified couplings, which is limited in our current approach. Specifically, our approach does not fully separate compounding or buffering effects where multiple drivers impact simultaneously. For example, circulation patterns identified as important may themselves be influenced by other factors not captured in our analysis. This potential for misattribution is an inherent limitation in our diagnostic framework and underscores the importance of interpreting our results as indicative of correlative relationships, rather than definitive causal links. Furthermore, identifying specific upper-level circulation patterns that drive SCF variability could build upon our findings for deeper insights. While our approach identifies the importance of circulation-related variables and their interactions across different reanalyses, a detailed assessment of circulation regimes would complement our statistical analyses with a more process-based understanding. It is also important to consider that estimates from interpretability frameworks within explainable ML (e.g., SHAP) are inherently dependent on the prediction of the ML model. Thus, employing a combination of different interpretability techniques, along with more generalizable and/or complex ML algorithms (e.g., deep neural/graph neural networks) can offer a more comprehensive understanding of Earth system interactions.

Our methodology primarily serves as a diagnostic tool to compare how different reanalysis products represent Earth system interactions, rather than to identify physical mechanisms. The value lies in (1) identifying biases across different models in representing non-linear interactions, which is crucial for investing in more computationally expensive process-based modeling; and (2) bridging the gap between process understanding and model evaluation by quantifying specific biases in the representation of interactions, beyond estimating general biases in the model outputs. We recognize that our approach is statistical in nature and does not directly identify physical mechanisms. While we can identify significant relationships and interactions among variables, establishing definitive causality requires detailed, process-based investigations. As such, our findings should be interpreted as highlighting potential relationships that can inform and guide subsequent detailed physics-based modeling efforts. As reanalyses and ESMs continue to evolve, our approach can support these development efforts by pinpointing the most critical coupling processes. Given the proof-of-concept nature of our study, applying similar analyses to other regions or phenomena could yield equally valuable insights into the model representation of Earth system interactions.

*Data availability.* Satellite and reanalysis data are available in the public domain. ERA5/ERA5-Land data were downloaded from the Copernicus Climate Data Store available at https://cds.climate.copernicus.eu/datasets?q=era5&limit=30. CAMS-EAC4 data were downloaded from the Atmospheric Data Store available at https://ads.atmosphere.copernicus.eu/datasets/cams-global-reanalysis-eac4?tab=overview. MERRA-2 and IMERG Final Run data were downloaded from NASA GES DISC available at https://disc.gsfc.nasa.gov/. MODIS SCF, LST, and MAIAC AOD were downloaded from NASA Earthdata https://search.earthdata.nasa.gov/search. The MATCHA dataset was recently re-

leased at the NSIDC (National Snow and Ice Data Center) for public use at https://nsidc.org/data/hma2_matcha/versions/1. An introductory article on MATCHA's model design is provided at https://himat.org/topic/matcha/. A GitHub repository at https://github.com/chayanroyc/Aerosol-Meteorology-Snow-HMA contains the necessary information on the methodology associated with this paper.

## Appendix A: Measures of Network Properties

### A1 Weighted Degree

For a network with $N$ nodes, the weighted degree $d_i$ of node $i$ can be mathematically represented as:

$$d_i = \sum_{i,j=1}^{N} w_{ij} \tag{A1}$$

where $w_{ij}$ is the interaction importance of the edge between node $i$ and node $j$.

### A2 Assortativity/Dissortativity

Assortativity (dissortativity) quantifies the tendency of nodes in a network to connect to other nodes with similar (assortative) or dissimilar (dissortative) importance, with values ranging from –1(perfect dissortativity) to +1(perfect assortativity). Positive values indicate that high-importance (high-degree) nodes preferentially connect with other high-importance nodes, whereas negative values indicate that high-importance nodes tend to connect with low-importance nodes. In our networks in (Fig. 4; Table 2), we see strong negative assortativity (i.e., high dissortativity). This arises because six aerosol (AER) nodes each link to 15 meteorological (MET) nodes, producing a clear hierarchy: primary/higher degree (AER) nodes connect more with secondary/lower degree (MET) nodes. Higher dissortativity therefore reflects greater heterogeneity in interaction importance, reflecting a complex hierarchical structure in AMI on snow across all networks.

The assortativity coefficient $r$ given by,

$$r = \frac{\sum_{jk} jk(e_{jk} - q_j q_k)}{\sigma_q^2} \tag{A2}$$

where $e_{jk}$ is the joint probability distribution of the node degrees, $q_j$ and $q_k$ are the remaining degree distributions, and $\sigma_q$ is the standard deviation of the distribution $q$. The joint probability distribution of the node degrees $e_{jk}$ is the probability that an edge connects nodes of degree $j$ and $k$, where the weighted degree is calculated is using Eq. (A1). Specific details on calculating network assortativity/dissortativity can be found in Newman (2002).

## Appendix B: Acronyms

**ACI** Aerosol-cloud interactions

**AER** Aerosol variables

**AMI** Aerosol-Meteorology interactions at the snow interface

**AOD** Aerosol optical depth at 550 nm

**ATPE** Adaptive Tree-Parzen estimators

**BC** black carbon

**BrC** brown carbon

**CCMM** Coupled chemistry meteorological model

**CLM-SNICAR** Community Land Model – Snow Ice Coupled with Aerosol and Radiation

**CMIP** Coupled Model Intercomparison Project

**DU** surface dust mixing ratio

**ECMWF** European Centre for Medium-Range Weather Forecasts

**ELEV** Elevation

**ES** Earth system

**ESP** Earth system predictability

**ESM** Earth system model

**GMTED** Global Multi-resolution Terrain Elevation Data

**GR** Glacial Regions

**HMA** High Mountain Asia

**HSC** High snow covered regions in High Mountain Asia

**IESM/IESA** Integrated Earth System Model/Analysis

**IMERG** Integrated Multi-satellitE Retrievals for GPM

**LAPs** light-absorbing particles

**LSC** Low snow covered regions in High Mountain Asia

**MAIAC** Multi-Angle Implementation of Atmospheric Correction

**MATCHA** Model for Atmospheric Transport and Chemistry in Asia

**MET** Meteorology variables

**ML** Machine learning

**MLR**  Multi-linear regression

**MODIS**  Moderate Resolution Imaging Spectroradiometer

**MOPITT**  Measurements of Pollution in the Troposphere

**NASA GMAO**  NASA Global Modeling and Assimilation Office

**NCAR**  National Center for Atmospheric Research

**PRECIP**  accumulated precipitation

**R22**  Roychoudhury et al., 2022 (Reference)

**RI**  Relative importance

**RGI**  Randolph Glacier inventory v6.0

**RRTMG**  Rapid Radiative Transfer Model for General Circulation Models

**SCF**  Snow cover fraction

**SHAP**  Shapley additive explanation

**SHAPc**  Shapley contribution

**WRF-Chem**  Weather Research and Forecasting Model with Chemistry

**XGBoost**  eXtreme Gradient Boosting

**PRECIP**  Daily accumulated precipitation

**QV2**  Specific humidity at 2 m

**ssHF**  Sensible heat flux at the surface

**ssLF**  Latent heat flex at surface

**T2**  Temperature at 2 m

**SKT**  Skin temperature

**T/H/M/L CC**  Total/high/medium/low cloud cover

**Z500/Z300**  Geopotential height at 500 hPa or 300 hPa

**MSLP**  Mean sea level pressure

**U10** Zonal wind speed at 10 m

**V10** Meridional wind speed at 10 m

**SS** Sea salt surface mixing ratio

**SU** Sulphate surface mixing ratio

**DU** Dust surface mixing ratio

**BC** Black carbon surface mixing ratio

**OM** Organic matter surface mixing ratio

$\tau$ Aerosol optical depth at 550 nm

*Author contributions.* CR contributed to the conceptualization, methodology, formal analysis, investigation, visualization, writing-original draft, reviewing, and editing. CH contributed to the funding acquisition, and writing - review and editing. RK contributed to the funding acquisition, and writing - review and editing. AFAJ contributed to the conceptualization, methodology, supervision, funding acquisition, and writing - review and editing.

*Competing interests.* The authors declare that they have no conflict of interest.

*Acknowledgements.* This work is supported by a NASA HiMAT2 grant (#NNH19ZDA001N-HMA). HiMAT2 is an interdisciplinary multi-investigator effort to understand the cryospheric and hydrological state of HMA. We also acknowledge the National Center for Atmospheric Research (NCAR) (sponsored by the National Science Foundation (NSF)) for assisting this ongoing study. This work is in tandem with the goals of the Aerosol subgroup under HiMAT2, to quantify the deposition of aerosols over snow in HMA. We would also like to thank Miguel Hilario and Debottama Das for discussions about this work.

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

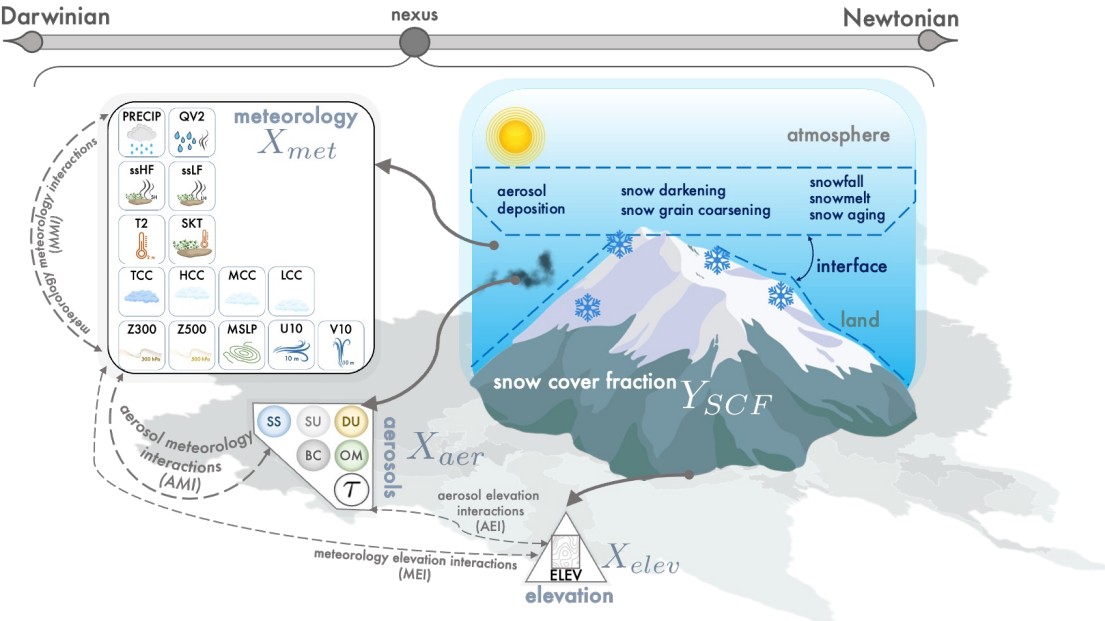

**Figure 1. Schematic describing an integrated approach to assess interactions at the atmospheric-cryospheric (land) interface over High Mountain Asia.** The Darwinian view focuses on the individual predictors (shown by the 22 icons grouped by meteorology, aerosols, and elevation) that drive snowmelt while the Newtonian view emphasizes emerging patterns and physics-based processes driving snowmelt (shown in the interface between the atmosphere and land). This study lies at the nexus of both perspectives where we assess the sensitivity of snow cover fraction to the interactions between aerosols and meteorology variables at the interface (aerosol-meteorology interactions or AMI on snow. Likewise, we also consider meteorology-meteorology interactions (MMI) on snow, aerosol–elevation interactions onto snow (AEI), and meteorology–elevation interactions (MEI) on snow. Abbreviations of the individual predictors are mentioned in the acronym section (Appendix B). The overlaid map of Asia is taken from FreeVectorMaps at https://freevectormaps.com/world-maps/asia/WRLD-AS-02-4001?ref=atr.

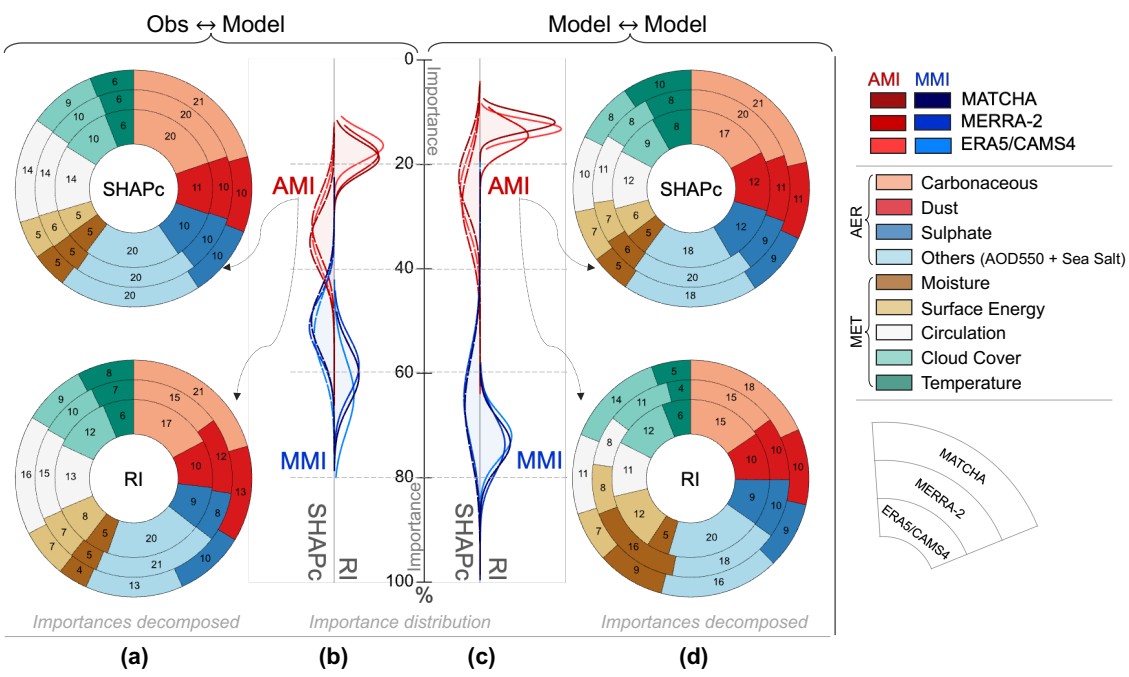

**Figure 2. Importance of aerosol-meteorology interactions on snow and their constituent variables.** Distributions of importance metrics (b-c), relative importance (RI), and Shapely contribution (SHAPc) for aerosol-meteorology (AMI) and meteorology-meteorology (MMI) interactions on snow shown for the Obs-Model (b) and Model-Model (c) construct for the three reanalyses. AMI's importance on snow is further decomposed into nine subgroups of predictors (four aerosol/AER and five meteorology/MET subgroups), which are shown in the donut pie-plots for the Obs-Model (a) and Model-Model (d) constructs for both RI (bottom row) and SHAPc (top row). The innermost ring shows the contribution of each subgroup to AMI's importance on snow from the ERA5-CAMS4 reanalysis, followed by MERRA-2 and MATCHA in the outermost ring.

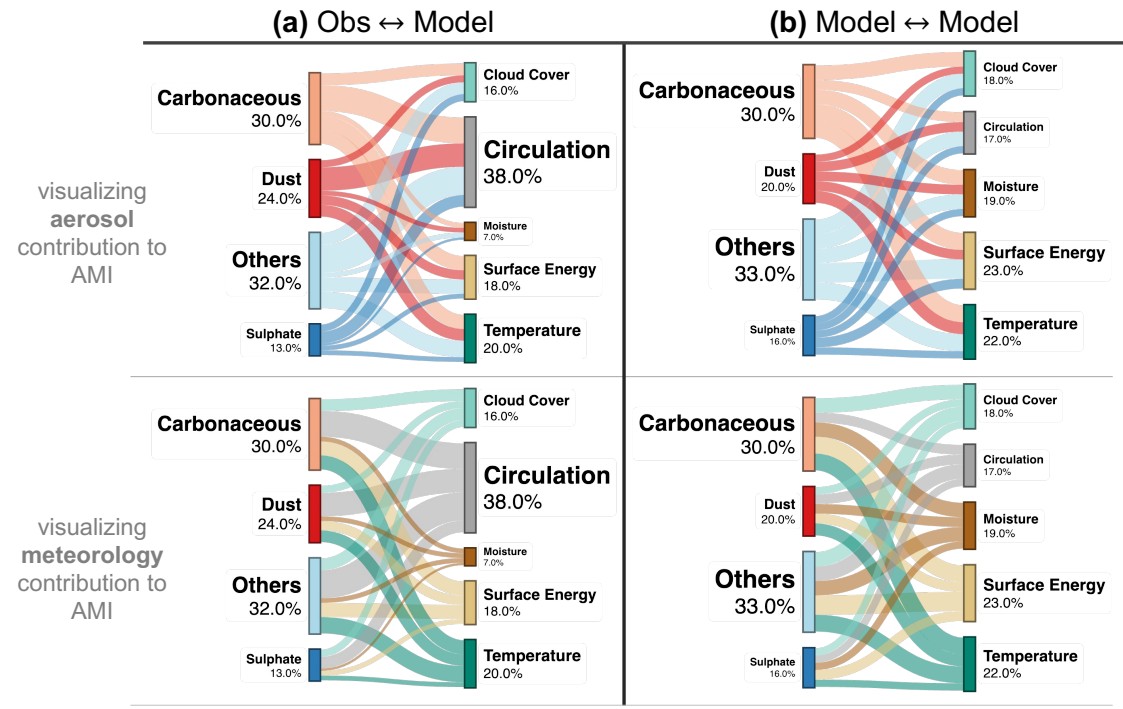

**Figure 3. Contribution of different aerosol and meteorology groups to importance of AMI on snow for both constructs across all three reanalyses.** Flow diagrams depicting the sum of the contribution of four aerosol groups and five meteorology groups to AMI in low snow-cover regions during the late snowmelt period (May-July) across the Obs-Model (a) and Model-Model construct (b), similar to the donut plots in Fig. 2(a) and 2(d), except the contribution is aggregated for all three reanalyses and both importance metrics. This contribution (also expressed in %) is the importance values ($\alpha$ in %) of the AER and MET predictors normalized to the total mean importance of AMI on snow. The top row shows the contributions color-coded by aerosol groups, and the bottom row shows the same contributions color-coded by meteorology groups. The *Others* aerosol group refers combined importance of both AOD at 550 and sea salt surface mixing ratio. The flow diagrams are made using SankeyMATIC.

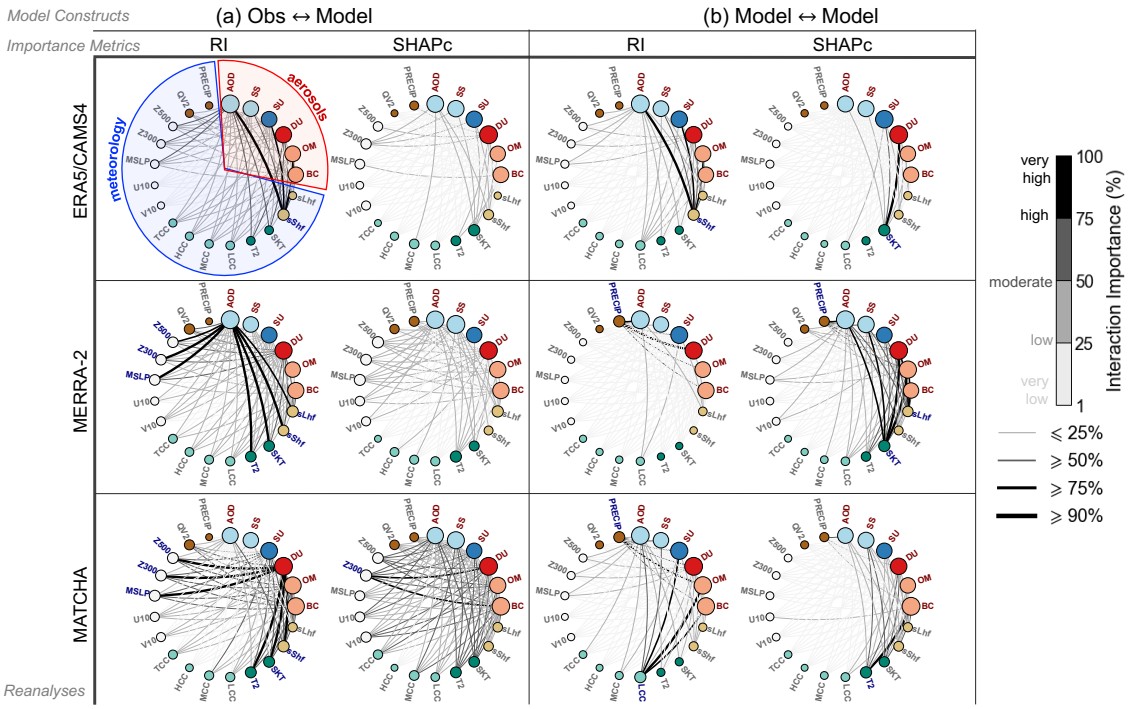

**Figure 4. Strength of aerosol-meteorology interactions on snow within each reanalysis for different metrics and constructs.** Network diagrams depicting the interaction importance/strength for both model constructs, Obs-Model (a) and Model-Model (b) using the relative importance (RI) and Shapely contribution (SHAPc) importance metrics of predictors within aerosol-meteorology interactions onto snow (AMI) in low snow-cover regions during the late snowmelt period (May-July). The nodes denote each predictor, while the lines (edges) denote the interaction importance on snow between the aerosol and meteorology variables, with their weights denoting the strength of the importance (1 to 100%, very low-low for <=25%, low-moderate for 25% to 50%, moderate-high for 50% to 75%, and high-very high for >=75% shown in the color bars). The node sizes differ between AER and MET predictors based on their weighted degree (number of edge connections, see Appendix A). The variable abbreviations at the nodes include the following aerosol variables; AOD for total AOD at 550 nm; SU for surface sulphate mixing ratio; SS for surface sea-salt mixing ratio; DU for surface dust mixing ratio; OM for surface organic matter mixing ratio and BC for surface black carbon mixing ratio. For meteorology, the abbreviations are as follows, PRECIP for daily accumulated precipitation; QV2 for specific humidity at 2 m; Z500 and Z300 for geopotential height at 500 and 300 hPa; U10 and V10 for zonal and meridional winds at 10 m; MSLP for mean sea level pressure; MCC, TCC, LCC and HCC for medium, total, low and high cloud cover fraction; T2 for temperature at 2 m; SKT for skin temperature; sShf and sLhf for surface sensible and latent heat flux; and finally ELEV for elevation. Details about these predictors can be found in Supplementary Table 2.

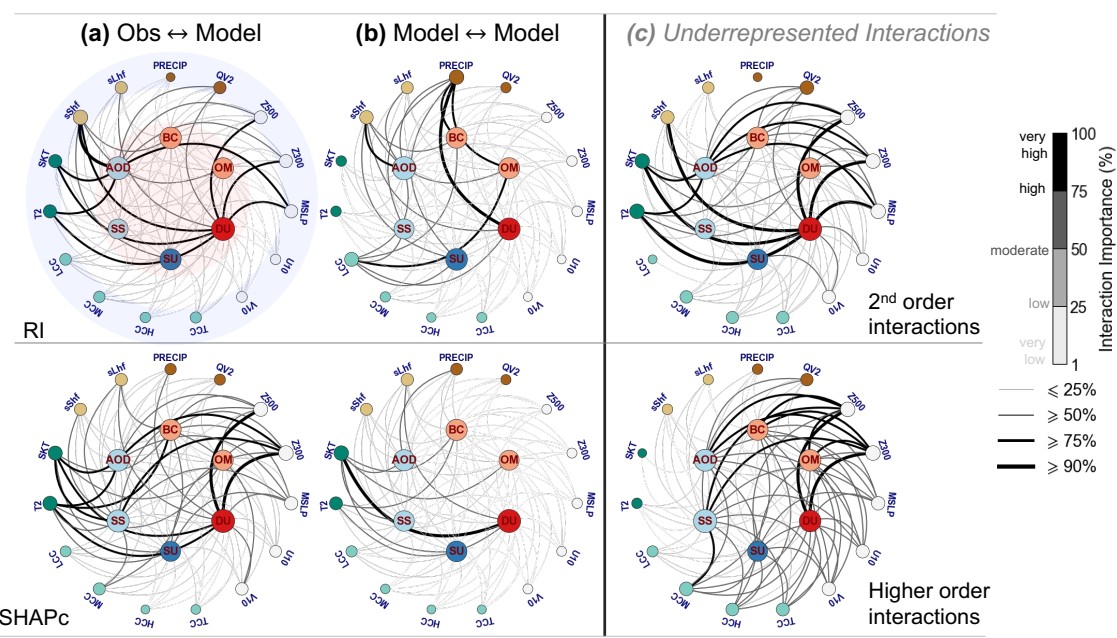

**Figure 5. Major aerosol-meteorology interactions at the snow interface for different importance metrics and constructs for all three reanalyses.** Network diagrams depicting the interactions aggregated across all three reanalyses (a-b) for each construct and importance metric (RI and SHAPc). Networks in (c) show underrepresented interactions captured by RI and SHAPc that should be emphasized across all three reanalyses. The nodes are arranged in a concentric fashion, with innermost nodes representing aerosol predictors (highlighted in red in the first network from top left) and the outermost nodes representing meteorology predictors (highlighted in blue in the first network from top left). The interaction importances are shown through edge connections between the nodes and are weighted by colors and width denoting the strength of the importance (1 to 100%, very low-low for <=25%, low-moderate for 25% to 50%, moderate-high for 50% to 75%, and high-very high for >=75% shown in the color bar).

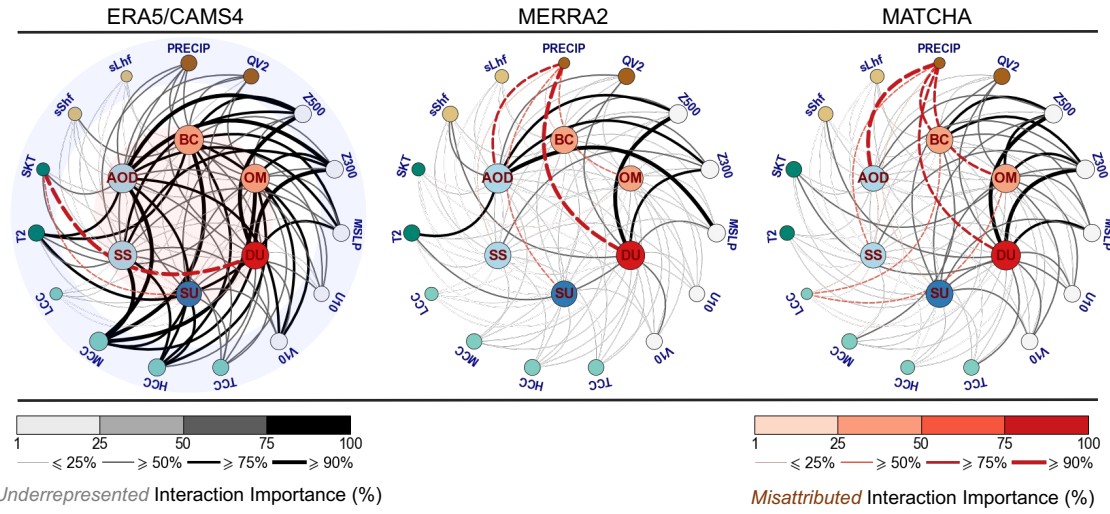

**Figure 6. Underrepresented and misattributed aerosol-meteorology interactions on snow in each reanalysis.** Network diagrams depicting the underrepresented and misattributed interaction importance/strength in AMI on snow across all three reanalyses. These are estimated using the difference in the interactions between the Obs-Model and Model-Model construct for each reanalysis. The positive differences shown by black edge connections highlight underrepresented interactions, while the negative difference shown by red edge (dashed) connections highlight misattributed interactions. The differences in the importance of these interactions (both positive and negative) are normalized separately (1 to 100%) for relative comparison. The nodes are arranged in a concentric fashion, with innermost nodes representing aerosol predictors (highlighted in red in the first network from left) and the outermost nodes representing meteorology predictors (highlighted in blue in the first network from left).

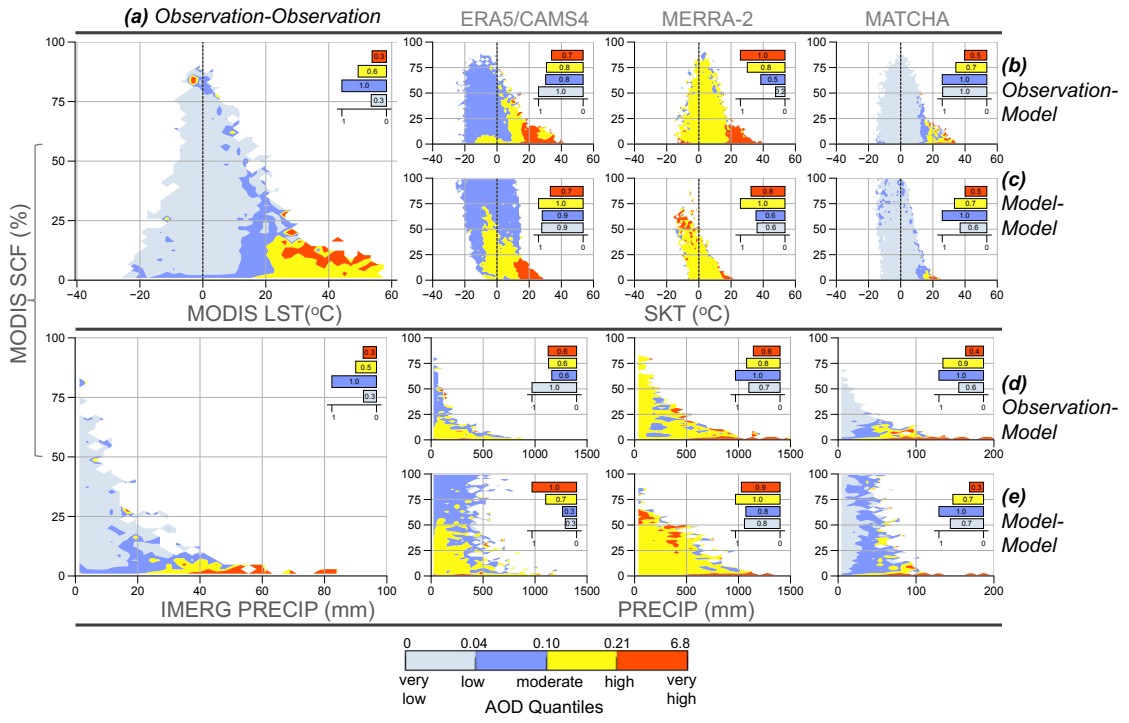

**Figure 7. Relationships between aerosols and meteorology to snow cover fraction in low snow-covered regions across all three constructs.** Scatter density of AOD at 550 nm based on snow cover fraction (SCF) in the y-axis with land surface temperature/skin temperature (LST/SKT) (top row) and daily accumulated precipitation (PRECIP) (bottom row) in the x-axes across the Obs-Obs construct (a), Obs-Model construct (b,d) and Model-Model construct (c,e). The variables are of daily resolution and masked for the low snow-covered regions. The scatter densities are classified (colored) by quantiles of AOD at 550 nm (shown in the legend) based on the respective aerosol datasets across the three constructs (MAIAC AOD for Obs-Obs, reanalysis AOD for the other two). The color bar represents the quantiles of AOD at 550 nm from very low to very high ($0^{th}$, $25^{th}$, $50^{th}$, $75^{th}$, and $100^{th}$ percentiles), computed across all three constructs (Obs-Obs, Obs-Mod, and the Mod-Mod construct). The bar graphs denote the non-linear sensitivity (quantified by max-normalized mutual information between 0 and 1) of SCF to LST/SKT and PRECIP at various quantiles of AOD for relative comparison of the sensitivity across different AOD quantiles.

**Table 1.** Statistics of the importance of AMI (in %) across all three reanalyses, two constructs and two importance metrics. $\mu$ refers to the average importance in % while $\sigma$ refers to the standard deviation of the importances in %.

| Construct | Obs-Model | | | | Model-Model | | | |
|---|---|---|---|---|---|---|---|---|
| | RI | | SHAPc | | RI | | SHAPc | |
| Reanalysis | $\mu$ | $\sigma$ | $\mu$ | $\sigma$ | $\mu$ | $\sigma$ | $\mu$ | $\sigma$ |
| ERA5/CAMS4 | 16.6 | 2.8 | 33.7 | 6.3 | 13.4 | 1.7 | 25.3 | 6.8 |
| MERRA-2 | 18.2 | 3.2 | 31.6 | 6.1 | 14.7 | 2.9 | 27.8 | 7.6 |
| MATCHA | 19.0 | 3.1 | 35.1 | 5.6 | 12.2 | 1.8 | 22.8 | 6.4 |

**Table 2.** Dissortativity based on the individual networks in Fig. 4. Dissortativity values range from 0 to -1. The values in parenthesis refer to the full range of dissortativity across using six importance definitions (using the mean, median, 5[th], 95[th], 25[th] and 75[th] percentile of the importance distributions for RI and SHAPc.

| | Dissortativity | | | |
|---|---|---|---|---|
| Construct | Obs-Model | | Model-Model | |
| Reanalysis | RI | SHAP | RI | SHAP |
| ERA5/CAMS4 | -0.8 (0.1) | -0.7 (0.0) | -0.6 (0.0) | -0.4 (0.0) |
| MERRA-2 | -0.5 (0.1) | -0.8 (0.0) | -0.4 (0.0) | -0.7 (0.1) |
| MATCHA | -0.6 (0.0) | -0.9 (0.0) | -0.7 (0.1) | -0.6 (0.1) |