# Peer review of "Diagnosing Aerosol-Meteorological Interactions on Snow within Earth System Models: A Proof-of-Concept Study over High Mountain Asia"

_EGUsphere, 2024_

## Author Response (AR1)

**Response for "Diagnosing Aerosol- Meteorological Interactions on Snow within the Earth System: A Proof-of-Concept Study over High Mountain Asia"**

**Response to Reviewer #1**

**Reviewer #1 General Comment 1:**

In this work, the authors use statistical and machine learning methods to assess how well reanalysis-derived aerosol, surface and meteorological predictor variables can predict MODIS-derived snow cover fraction in high mountain Asia. They focus on the relative importance of different non-linear interaction terms between the predictors and snowmelt during May-June. From this information, they make suggestions on which aerosol/meteorological interaction processes models can improve on to better model snow-related processes in this region.

This seems like a very interesting approach that really could be useful to other applications, as the authors suggest. Their method can provide clues on what processes to prioritize for future study, either in models or with field studies. I found the scientific results to be very interesting, assuming they didn't misattribute something due to noise. However, while they discuss uncertainties broadly in section 4.3, I do believe that they need a more complete discussion of the uncertainties in the paper when they discuss individual findings. Also, while my impression of the quality of the work is generally favorable, I am not an expert in machine learning methods and so it would be helpful to have the complementary opinions of another reviewer with that expertise.

Otherwise, the figures are attractive and fairly easy to understand. It is well-written overall, but the introduction seemed too theoretical and could be made more succinct to encourage readers to really grasp what is new and important in this study. The abstract also doesn't quite mirror section 4 in terms of the implications of the study. Also, while I was impressed with the paper's thoroughness in some places, the methods are incompletely described in some places (see more details below), such that I believe that the study would not be easy to replicate as currently written.

**Authors' Response:**

We sincerely thank you for your comments on our manuscript. We appreciate the positive feedback on the scientific value of our approach and your recognition of its potential applications. We have addressed your specific concerns in the sections below. We are grateful for your constructive feedback, which has helped us improve the clarity and rigor of our manuscript. In particular,

1. We have added to our discussion of uncertainties and implications to sensitivity throughout the manuscript, particularly in the data sources within Methods (Section 2), when discussing individual findings by mentioning previous studies (Section 3) and in the Limitations and

Future Directions section (Section 4.3). Please see our responses related to the uncertainties in the Specific comments below.

2. We have revised the introduction for more clarity and shortened it by removing redundant/obtuse statements and arranged it in a coherent manner that reflects the novelty, importance, and objectives of our study. We have added an outline of the manuscript as well at the end of the introduction for clarity.

3. We have added to our discussion of uncertainties throughout the manuscript, particularly in the data sources within Methods (Section 2), when discussing individual findings by mentioning previous studies (Section 3) and in the Limitations and Future Directions section (Section 4.3).

4. We have revised the abstract to better reflect the implications discussed in Section 4.2. We have modified the last few lines of the abstract to reflect this as follows,

> *These insights point to the degree of complexity of these interactions and their relative strength of representation across ESMs. The proposed framework can be extended to help diagnose other complex Earth system processes and complement conventional feedback separation methods. This has broader implications for the future development of coupled models to improve Earth system predictability.*

5. We have significantly expanded the methods section using Supplementary information with additional details on the machine learning model configuration to ensure reproducibility, including more explicit descriptions of our machine learning implementation and statistical procedures.

6. We have revised our title to *Diagnosing Aerosol-Meteorological Interactions on Snow within Earth System Models: A Proof-of-Concept Study over High Mountain Asia* to emphasize that our study focuses on diagnosing the differences in the representation of interactions relevant to snowmelt across three reanalyses (and the inherent model design) rather than identifying observable interactions or physical pathways.

More discussion of the uncertainties is merited:
Please provide information on how good the predictors (e.g., dust) are in the reanalyses. Has there been any validation for these predictors in this region? It may be worth noting in the text that the mountainous terrain can make these values less certain. It is also important to more clearly note that a low predictability could either be due to low importance of the factor or poor reanalysis skill of the factor.

**Authors' Response:**

Thank you for raising this important point about the skill of the predictors, especially for complex terrain. ERA5/CAMS-EAC4 and MERRA-2 are some of the widely used and extensively validated reanalyses and detailed evaluations are present in their individual papers (references in Section 2.1). While we acknowledge that a detailed evaluation of all the variables is needed, it is outside the primary focus of this manuscript. We have modified the manuscript in the following sections to mention the biases within the reanalyses, the predictors, and their implications on the sensitivity, as follows,

1. We added a few studies to discuss concisely the biases within the reanalyses in the Methods section for reanalyses (Section 2.1) as follows,

   *ERA5 exhibits systematic wet and warm biases over Asia, with higher near-surface wind speeds (Sun, 2017; Gong et al.,2022; Wei et al., 2024). CAMS-EAC4 captures large-scale aerosol transport but underestimates total and speciated aerosol concentrations, particularly during high aerosol events, while overestimating BC. MERRA-2 generally simulates higher dust concentrations and better represents extreme aerosol events, but also overestimates BC (Li et al., 2024; Ansari and Ramachandran, 2024; Gueymard and Yang, 2020; Xian et al., 2024). It is also important to note that while these reanalyses are invaluable for studying regions with sparse observations such as HMA, their application in high-elevation regions has its challenges. Several studies in the region have pointed out elevation biases in surface temperature and its trends, as well as wind patterns arising from complex topography, varying vegetation cover, and the coarser resolution of reanalyses, which cannot resolve valley-scale terrain (Luo et al., 2019; Tang et al., 2022; Jentsch and Weidinger, 2022; Pepin and Seidel, 2005).*

2. In the Limitations and Future Directions (Section 4.3) we added,

   *Moreover, reanalysis datasets are known to exhibit biases, particularly in high-elevation regions (see Section 2.1); hence, their relative skill and accuracy must be taken into account when interpreting their sensitivity to the target variable.*

3. Section 3.4 (Observable AMI on snow) analyzes relationships between the observed target

(SCF) from MODIS, and observed and modeled predictors (temperature, precipitation, and aerosol optical depth at 550 nm) across the three reanalyses, which highlights their skill in representing these quantities over HMA. We can also see from Figure 7, that these variables don't differ significantly across the reanalyses as well.

4. We show in Supplementary Figures 1 and 4 the spatial and temporal average (2003-2019) of some variables relevant to our findings: snow cover fraction, dust, and black carbon mixing ratio over the study region and period.

5. 1. We discuss Supplementary Figure 1 and the differences in snow cover fraction across the reanalyses at the beginning of Results (Section 3).

6. The evaluation for MATCHA is currently in preparation. We show the spatial and temporal distribution of some predictors within MATCHA in Supplementary Figures 1 and 4.

7. Regarding the implications on the sensitivity (importance of the factors), we employed several methods to ensure robustness as follows,

- We analyzed data from three different reanalysis products, which allowed us to identify consistent patterns across multiple models.
- We utilized two fundamentally different algorithms for calculating importance metrics, one statistical and one machine learning-based, which reduces the likelihood that our findings are artifacts of algorithms.
- We compared the networks across the Obs-Model and the Model-Model construct, to distinguish between what we see through observations and models.

While these approaches increase confidence in our findings, we acknowledge that a comprehensive uncertainty quantification would require observational datasets for all predictor variables, which still remains a challenge. We mention this in the revised manuscript within the Limitations and Future Directions (Section 4.3) as follows,

*Observational datasets for predictors that are currently derived primarily from reanalyses are also needed to conduct a more comprehensive analysis, beyond what we presented in Section 3.4, where we evaluated the joint distributions of interacting variables in relation to the target variable.*

**Reviewer #1 Specific Comment 3:**

Relatedly, in Table 2: Please discuss the role of data noise in the results from this table and implications for the interpretation of it.

**Authors' Response:** Thank you for highlighting the role of data noise in the dissortativity of the networks. The dissortativity values and the network visualizations were based on the mean of the importance distributions shown in Figure 2. We did a preliminary analysis of the changes in dissortativity by using six importance definitions (mean, median, $5^{th}$, $95^{th}$, $25^{th}$ and $75^{th}$ percentile) for the importance distributions for RI and SHAPc. We found that the dissortativity changes by a maximum of 0.1 across these importance definitions. We have now modified our Table 2 to show these ranges. We also see that some of the dissortativity values do not show any change.

Regarding the potential influence of data noise on the dissortativity values, we employed several methods to ensure robustness as follows,

- We analyzed data from three different reanalysis products, which allowed us to identify consistent patterns across multiple models.

- We utilized two fundamentally different algorithms for calculating importance metrics, one statistical and one machine learning-based, which reduces the likelihood that our findings are artifacts of algorithms.

- We compared the networks across the Obs-Model and the Model-Model construct, to distinguish between what we see through observations and models.

While these approaches increase confidence in our findings, we acknowledge that comprehensive uncertainty quantification would require

- observational datasets for all predictor variables, which remains a challenge,

- verifying these networks across the distribution of the importance values which is beyond the scope of this proof-of-concept.

We mention this in the revised manuscript within the Limitations and Future Directions (Section 4.3) as follows,

> *In addition, exploring the uncertainty within the networks would be valuable, allowing us to quantify the confidence in the inferences drawn in this study. Observational datasets for predictors that are currently derived primarily from reanalyses are also needed to conduct a more comprehensive analysis, beyond what we presented in Section 3.4, where we evaluated the joint distributions of interacting variables in relation to the target variable.*

We also want to mention that the consistency of higher dissortativity values in Table 2, and their differences between the constructs (at most 0.3) compared to the uncertainty range of 0.1 suggests

that the values are not solely a product of noise. We have included this text in Section 3.3.1 to reflect this as follows,

*It is important to note here that the network dissortativity also indirectly depends on how well the predictors, and their interactions, capture sensitivity to the target variable (SCF). This affects the importance metrics (RI and SHAPc), which in turn influence the network edges used to calculate dissortativity. Errors in reanalysis products, which serve as our observational proxies, can potentially manifest as stronger or weaker connections in the networks, thus affecting the dissortativity. However, the consistency of higher dissortativity values across different reanalyses and importance metrics (from Table 2), along with the maximum uncertainty range of 0.1 (shown in parentheses), suggests that the observed differences primarily reflect real structural limitations in how models capture the hierarchical complexity of aerosol–meteorology interactions on snow, rather than mere statistical artifacts. Furthermore, the dissortativity differences (up to 0.3) between Obs-Model and Model-Model constructs indicate that these differences represent meaningful structural variations in aerosol-meteorology interactions that exceed what would be expected from random fluctuations in the data due to limited samples.*

**Reviewer #1 Specific Comment 4:**
Fig. 4: In the text, it might be helpful to discuss the implications on the results of missing/not including any key variables in the analysis.

**Authors' Response:**
Thank you for pointing out the need to discuss missing variables. We expanded the text accompanying Figure 4 to highlight the potential implications of missing or excluded variables, such as biases introduced by incomplete datasets. We have included this text in Section 3.3 to reflect this as follows,

*It is important to note here that the network edges that represent the strength of the interaction are based on the importance metrics (RI and SHAPc) that are calculated based on the 21 aerosol and meteorological predictors. As such, considering a different set of predictors might influence these importance values and the network edges. We accounted for this possibility by selecting a diverse range of 22 predictors across nine groups of variables (five meteorological, four aerosol) which we show in Figure 3.*

We also mention in Limitations and Future Directions (Section 4.3),

*We acknowledge that the insights from our approach are dependent on the choice of variables to represent these processes. Thus, future studies incorporating other relevant variables (e.g., net surface radiation fluxes, boundary layer height, vertical profiles of thermodynamic variables,*

*and wind) will also be valuable.*

Figure 6: I'd like to see a more detailed discussion of possible uncertainties in this figure.

**Authors' Response:**

Thank you for mentioning this. We revised our manuscript to discuss the uncertainties in Figure 6, focusing on potential misrepresentations of second-order interactions and the limitations of reanalysis frameworks in capturing higher-order interactions as follows, in Section 3.3.3.

*While these aggregated networks (in Fig. 6) can highlight the higher-order interactions within each reanalysis, it is necessary to consider the skill of these datasets in accurately depicting real-world interactions, which may be limited by the absence of parameterizations in their respective model frameworks to represent these interactions, or by misrepresentation of the order of interactions between these predictors (second order or higher) if at all present within the reanalysis frameworks.*

We also want to emphasize that quantifying detailed uncertainties within the networks would require verifying these networks across the distribution of the importance values which is beyond the scope of this proof-of-concept. We revised the Limitations and Future Directions (Section 4.3) to reflect this,

*In addition, exploring the uncertainty within the networks would be valuable, allowing us to quantify the confidence in the inferences drawn in this study.*

**Reviewer #1 Specific Comment 6:**

Section 3.3.4. Very interesting scientific results, assuming the authors didn't misattribute something due to noise in the data. Authors, please address this point in the text.

**Authors' Response:**

Thank you for the interest in the results and your concern about potential misattribution. We have addressed the biases within the predictors and the networks and elaborated them in Specific Comments 2,3 and 5. We have also added a text in Section 3.3.4 to reflect this as follows,

*As mentioned previously, the interpretation of misattribution of these interactions within each reanalysis depends on the accuracy of these predictors in representing their sensitivity to SCF, as well as the representation of these feedbacks within the reanalysis frameworks.*

**Reviewer #1 Specific Comment 7:**

Clarification on the methods is also requested:
Section 2: Please explain where were these reanalysis output data were taken from? Near the site? Over the site?

**Authors' Response:**

Thank you for requesting clarification on this. We have used the model surface for the aerosol mixing ratios and the meteorological variables unless mentioned in Supplementary Table 1 (such as geopotential height at 300 hPa). The reanalysis outputs are not of sites but were regridded to 0.75° horizontal resolution over each of the glacier regions from the RandolpInventory inventory v6 (six regions which are mentioned in Section 2.4 and caption of Supplementary Figure 1.).

**Reviewer #1 Specific Comment 8:**

- Discussion on Equation 1: The code for the project is not publicly available yet, and so more detail needs to be provided here for others to replicate this work. For example, what software and method/packages were used in this part of the analysis? Did the authors take steps against over-fitting the data?

**Authors' Response:**

Thank you for mentioning this point. While the code for the importance algorithms is not publicly available yet, the entire algorithm for calculating relative importance is mentioned as supplementary information of the reference R22 (Roychoudhury et al., 2022, mentioned below). The algorithm for using SHAPc is also mentioned in Section 2.7 and the Supplementary Text as explicitly as possible. The packages used for the machine learning model and SHAPc are mentioned in the Supplementary as well. We primarily used Python for these importance algorithms, networks, and regression and mention specific details of our machine learning methods in the Supplementary Text for reproduction. For network visualizations, we have used the `networkx` package (v2.8.4) and Gephi (v0.10) for the network graph layouts, which we have added in the manuscript. All components of our approach are already established using open-source packages which we have properly cited across the manuscript.

We accounted for the possibility of over-ftting by optimizing for hyper-parameter tuning of the XGBoost algorithm (the adaptive Tree-Parzen estimator method in the hyperopt package as mentioned in Section 2.7) in a step-wise manner. We used the XGBoost to fit over 5 folds (subsets of the data) and then fitted it to the entire dataset before optimization of hyperparameters. The goal was to find appropriate hyper-parameters after sufficient iterations that provide the best fit for the regression, rather than focus on the predictability of the target variable for which over-fitting could be an issue. The idea was that given sufficient depth within the regression (which is itself a hyper-parameter), the XGBoost algorithm would be able to represent the necessary higher-order interactions between the predictors after optimizing the hyper-parameters, rather than focusing on

the generalizability of the model. We have however, mentioned this as a limitation and a future direction in our Limitations and Future Directions and Future Directions (Section 4.3). Our results on the similarity of the conclusions between the statistical (relative importance) and the machine learning (SHAPc) metrics about the importance of AMI on snow cover fractions provided evidence to back this idea, due to which we did not pursue the idea of generalization further. We mention this limitation in Section 4.3 (Limitations and Future Directions).

*Roychoudhury, C., He, C., Kumar, R., McKinnon, J. M., and Arellano Jr., A. F.: On the Relevance of Aerosols to Snow Cover Variability Over High Mountain Asia, Geophysical Research Letters, 49, e2022GL099 317, 2022*

**Reviewer #1 Specific Comment 9:**

L680: "The codes for MATCHA's model framework, the regression algorithms, and the analysis will be made available in a public repository after publication." It would be better if these were available for the publication. Without that, or a much more detailed description of the methods, I don't think people will be able to replicate these results, as is the standard for most journal articles.

**Authors' Response:**

Thank you for highlighting the importance of code availability. We have revised our manuscript to include methodological details in the Supplementary and the text to ensure replicability (also elaborated in the previous comment). We describe the regression algorithms in the Methods section, the model configuration for the machine learning model, as well as calculation of SHAPc values in the Supplementary. The entire algorithm for calculating relative importance from multiple linear regression is mentioned in the supplementary information of the reference R22 (Roychoudhury et al., 2022). We have also mentioned a link to NASA's Snow and Ice Data Center which hosts the entire MATCHA simulation (`https://nsidc.org/data/hma2_matcha/versions/1`). The code for MATCHA simulations is based on modifications in the default WRF-Chem code from NSF NCAR (v3.9) which is available upon request and made public after the submissions of the paper evaluating MATCHA. The following has been added to our data availability section,

*The MATCHA dataset is recently released at the NSIDC (National Snow and Ice Data Center) for public use at `https://nsidc.org/data/hma2_matcha/versions/1`. An introductory article on MATCHA's model design is provided at `https://himat.org/topic/matcha/`.*

- L224. "Both metrics are calculated to add up to 100%, making it easier to interpret these sensitivities." This is actually not very easy to interpret, at least not for me. For example, in Fig. 3, it looks like carbonaceous aerosols predict ∼25% of the variability in circulation, but that is not actually correct, as I understand it. I think more explanation here and, in the text, discussing Fig. 3 is necessary for readers to not accidentally be misled by this approach.

**Authors' Response:**

Thank you for noting the need for clarity regarding the percentage values of the metrics. We want to emphasize that we calculated both relative importance (RI) and SHAPc as percentages from their absolute values obtained from the regression results, which are represented as $\alpha$ in Eq.1 and 2 and used in Figures 2b and 2c (shown in the probability distribution). The closest analogy to this is obtaining regression slopes of predictors in traditional multiple linear regressions (which represent the sensitivity of the predictors to the target) and representing these slopes as percentages. The importance values of AMI or MMI, as shown in Figure 2b/2c as the sum of $\alpha$ (which is in %) that correspond to the product terms containing both aerosol and meteorological predictors. These percentage values are then used for interpretation of the results in Section 3.1. The percentage values in Figure 3 are instead not the importance values $\alpha$, but normalized values of these importance percentages to the total importance of AMI on snow (that is based on $\alpha$). We discuss in detail the normalizing of the SHAP values to percentages in the Supplementary. We have mentioned this clarification in the text to avoid further confusion on the percentage values.

*This contribution (also expressed in %) is the importance values ($\alpha$ in %) of the AER and MET predictors normalized to the total mean importance of AMI on snow as mentioned before.*

- L241: "non-linear interaction terms defined as product terms between these predictors (253 in total)" Please provide more information here. As written, it is not clear what the 253 predictors are, how they were derived, and what they are meant to represent. This was somewhat further explained in section 2.7, but details are still lacking. For example, it would be helpful to explain where the 231 value comes from (Line 274). Please also define a "pairwise interaction contribution."

**Authors' Response:**

Thank you for pointing out the need to clarify where these numbers come from. As mentioned in Section 2.6, we have 22 predictors in total (six aerosol predictors, 15 meteorological predictors, and an elevation variable) which represent the main effects in the multiple linear regression. The interaction effects are represented by their product terms (excluding square terms), which amount to 231 product terms, bringing the total number of predictors to $22 + 231 = 253$ This is calculated using combinations, $^{22}C_2$ or $\binom{22}{2}$, which amounts to 231 product terms in total. For example, with three predictors, you would $^{3}C_2 = 3$ product terms. Thus, there would be 6 terms in the predictor

side of the regression.

Regarding "pairwise interaction contribution", we clarify this in two places, Line 245 where we mention,

> *where the dependence ($\alpha$) on a predictor $X_i$ is not a constant, but dependent on a second predictor $X_j$.*

and Line 227, where we mention,

> *Note that the metrics of importance are bivariate, reflecting the joint sensitivity of SCF to a predictor in the presence of another predictor. Importance of AMI on snow can thus be interpreted as the impact of MET predictors on snow in the presence of AER variables.*

**Reviewer #1 Specific Comment 12:**

Other specific comments:
L583: "remain the primary driver of SCF ( 60% contribution)" I don't think the authors can say that 60% of the driving factors are MMI. What they can say is that 60% of the predictable variability in SCF comes from MMI in their analysis. It would be helpful to clarify this throughout section 4.

**Authors' Response:**

Thank you for pointing out the potential overstatement. We have now modified this sentence as follows,

> *While interactions within meteorology at the snow interface (MMI) contribute the most to the variability of SCF ($\sim$60% contribution), drivers related to AMI account for an average 20% of the SCF variability.*

**Reviewer #1 Specific Comment 13:**

Section 4: The confidence in their methods in this section seems a bit overblown. It is not that I disagree too much with the main points, but I feel that greater emphasis on the uncertainties would be appropriate. For example, on L608: the authors say, "Although both MERRA-2 and MATCHA incorporate some degree of coupling within their models, interactions of dust with circulation variables need more attention within the two." I would instead have said, "Although both MERRA-2 and MATCHA incorporate some degree of coupling within their models, our study suggests that interactions of dust with circulation variables need more attention within the two."

**Authors' Response:**

Thank you for pointing this out. We have modified the text in Section 4 as recommended to reflect

that the results arise from our interpretation of the results,

> *Although both MERRA-2 and MATCHA incorporate some degree of coupling within their models, our study suggests that interactions of dust with circulation variables would need more attention within the two. The models in both these reanalyses seem to overemphasize interactions of aerosols (particularly dust) with daily accumulation precipitation, instead of coupling with circulation variables such as geopotential height and mean sea level pressure. From our interpretation of the networks, it seems that ERA5/CAMS4 relies extensively on its non-coupled model framework and assimilation of observations and needs extra attention to circulation and cloud cover-related interactions in the future development of the ECMWF model. The variability in the importance distribution of AMI on snow across the reanalyses is also lower than the difference in the variability of AMI's importance on snow from both constructs (Fig. 2 and Table 1) indicating that the coupling within MATCHA is far from ideal. Thus, the need for parameterizations that represent the feedbacks between snow and aerosol abundances, including relevant snowmelt drivers like circulation-related variables is necessary to consider in the development of future ESMs.*

We have also mentioned the uncertainties inherent in our findings and inferences throughout the revised manuscript as mentioned in the specific comments previously.

**Reviewer #1 Specific Comment 14:**

L201. It would be helpful to refer the reader to Supplemental Table 2 when mentioning the predictors.

**Authors' Response:**

Thank you for this comment. We have modified our text to reflect this as follows,

> *A total of 22 variables (six aerosol, and 15 meteorology-related) from the three reanalysis datasets in addition to elevation, were selected as predictors that can potentially drive SCF (see Supplementary Table 2 for the predictors).*

**Reviewer #1 Specific Comment 15:**

Supplementary Fig. 1a: These figures are confusing because it is hard for a reader to orient in terms of location and area. I recommend adding information on latitude, longitude and a length scale bar. Fig. 1b is very good.

**Authors' Response:**

Thank you for this feedback. We have updated Supplementary Figures 1a and 4a to display the coordinates and length scale.

**Reviewer #1 Specific Comment 16:**

Intro: might be good to mention the fraction of the planet that is covered by the Third Pole for context

**Authors' Response:**

Thank you for this comment. We have mentioned the fraction of the planet covered by the Third Pole for context (5 million sq. km or 1%) in the introduction as follows,

> *The rapid acceleration of glacial snowmelt in recent decades has critically impacted the freshwater resources that support the livelihood of regions downstream of the glaciers in High Mountain Asia (HMA), often referred to as the Third Pole, which contains the world's largest reservoir of glaciers and snow ($\sim$1% of the Earth's surface area) outside of the Earth's polar ice sheets (Refs).*

**Reviewer #1 Specific Comment 17:**

It would be helpful to define "network theory" in the abstract when it is first discussed since this is not a term many in the field will be familiar with.

**Authors' Response:**

Thank you for this comment. We have modified the abstract to include a short definition of network theory as follows,

> *Here, we use network theory, a graphical approach that maps the relationships between variables as interconnected nodes, to identify key variables that influence snowmelt processes.*

**Reviewer #1 Specific Comment 18:**

L240: what is C2? please define.

**Authors' Response:**

Thank you for this comment. Here, we used the combinatorial symbol to calculate combinations of 2 samples out of 22 predictors. We have replaced the $^{22}C_2$ in the text with a more conventional form $\binom{22}{2}$ to represent "22 choose 2".

**Reviewer #1 Specific Comment 19:**

L 448: "in MERRA-2, second-order interactions between DU and AOD with circulation and temperature variables are absent" Please describe some examples of such potentially important second-order interactions

**Authors' Response:**

Thank you for this comment. Here are two examples of these second-order interactions that we consider to be a component of AMI (aerosol-meteorology interactions) on snow,

1. Dust/black carbon/AOD with geopotential height. Absorbing aerosols can modify atmospheric circulation by warming the atmospheric layers (increase in geopotential height) which can lead to convection patterns and influence regional temperature and precipitation. This in turn can influence snow cover.

2. Black carbon with near-surface temperature (at 2 m). Black carbon deposition on snow reduces snow albedo (a part of aerosol-induced snow albedo feedback) which leads to enhanced absorption of solar radiation and increase near-surface warming, which can accelerate snowmelt.

We have included these two examples in our revised introduction (end of paragraph 4) as follows,

> *Examples of these second-order interactions can include (a) interactions between absorbing aerosols and geopotential height that can modify convection patterns, regional temperature, and precipitation and (b) interactions between BC and near-surface temperature that reduce snow albedo through near-surface warming.*

**Reviewer #1 Specific Comment 20:**

L. 501: "We see that interactions between DU and PRECIP in MERRA-2 and MATCHA are given unnecessary importance, while for ERA5/CAMS4, it is the interactions between DU and SKT that are overemphasized." Is there any supporting evidence for this hypothesis in the literature?

**Authors' Response:**

Thank you for mentioning this. While there is no direct evidence that we have yet found that mentions this specifically about ERA5 or MERRA-2 reanalyses, below are a few studies that can support the idea of what we mentioned in Line 501,

- Pu and Ginoux, (2018) mention the underestimation of CMIP5 dust simulations and their overestimation of the impact of surface wind and precipitation on dust abundance within models.

- Kok et al., 2017 mentions how global climate models underestimate the abundance of coarser dust particles, and hence leading to incorrect estimates of dust-radiative feedbacks.

- Stante et al., 2023 mentions how high concentrations of dust aerosols can influence biases between satellite and land surface temperatures and ERA5's SKT over the Sahara region. This can also be a consequence of the fact that turbulent heat fluxes near the surface are not well parameterized, or that boundary layer height is not accurately adjusted due to warming by dust aerosols.

- Zhao et al., 2018 mention large uncertainties and inter-model diversity in simulating dust aerosols across CMIP6 models. They also mention how these models fail to reproduce the dust distribution in the southern edges of the Himalayas. They mention that modeled dust processes are becoming more uncertain as models become more sophisticated.

We have added these references and a sentence to support our hypothesis in our manuscript around Line 501 as follows,

*Previous studies have identified significant biases in model representations of dust processes, such as overestimation of precipitation's impact on dust abundance as well as large variability in dust simulations across models (Pu and Ginoux, 2018; Kok et al., 2017; Zhao et al., 2022 ), and the complex interactions between dust aerosols and surface temperature that can lead to biased parameterizations of near-surface processes (Stante et al., 2023).*

*References*

- *Pu, B., & Ginoux, P. (2018). How reliable are CMIP5 models in simulating dust optical depth? Atmospheric Chemistry and Physics, 18(15), 12491–12510.*

- *Kok, J. F., et al. (2017). Smaller desert dust cooling effect estimated from analysis of dust size and abundance. Nature Geoscience, 10(4), 274–278.*

- *F. Stante, S. L. Ermida, C. C. DaCamara, F. -M. Göttsche and I. F. Trigo, Impact of High Concentrations of Saharan Dust Aerosols on Infrared-Based Land Surface Temperature Products. in IEEE Journal of Selected Topics in Applied Earth Observations and Remote Sensing, vol. 16, pp. 4064-4079, 2023, doi: 10.1109/JSTARS.2023.3263374.*

- *Zhao, A., C. L. Ryder, and L. J. Wilcox, 2022: How well do the CMIP6 models simulate dust aerosols? Atmospheric Chemistry and Physics, 22, 2095–2119, https://doi.org/10.5194/acp-22-2095-2022.*

**Reviewer #1 Specific Comment 21:**

L580: "We substantiated the importance of AMI on snow in driving SCF variability..." The wording here seems a bit strong. To substantiate the importance of something to me means that you are proving something, but what I think the authors meant (and what I think it more accurate) is that they provided additional information in support of this importance.

**Authors' Response:**

Thank you for the comment. We understand that the use of the word "substantiated" can imply a direct proof of our statement, so we instead revised it as follows,

> *We estimated the importance of AMI on snow in driving SCF variability across three reanalyses and two importance metrics during the late snowmelt season, building on our previous work (Roychoudhury et al., 2022).*

**Reviewer #1 Specific Comment 22:**

Section 4.2: In addition, in principle, if the method is as accurate as they think it is, could be used to help focus field campaigns to get the most bang-for-the-buck. But it would be good to get supporting literature evidence for their approach working first, as they mention in Section 4.3.

**Authors' Response:**

Thank you for highlighting this potential application. Our approach in this study is an attempt to identify the key variables and their interactions that are important for snow cover fraction, which can potentially be leveraged for field campaigns by prioritizing the design of observational sites where stronger interactions can be estimated, where reanalysis datasets diverge significantly (needing targeted measurements), as well as identify high-impact processes that might be under-represented in current reanalyses. We incorporated your comment in our Limitations and Future Directions (Section 4.3) to urge further studies using this approach for more supporting evidence as follows,

> *Beyond improving ESMs, our methodology can also inform a more optimal design of field campaigns in climate-vulnerable regions such as HMA. By identifying the key variables and interactions that affect snowmelt (or any other phenomenon of interest), and determining when and where these interactions are more pronounced, our approach can help optimize the allocation of limited observational resources. This would enable more strategic selection of the variables, locations, and periods to yield the most valuable information on critical Earth system processes in these regions.*

**Reviewer #1 Specific Comment 23:**

Great that they have supplementary table 1. That's a good resource and important to have available. Section 4.3 is really great, and I am glad they added that in. Sentence starting on L. 360: this is a really important point, and I am glad the authors included it.

**Authors' Response:**

Thank you for these comments. As we used around 21 predictors, a table (Supplementary Table 1) would be the best way to mention all the necessary variables used for our study. The limitations in Section 4.3ares an attempt to address all possible shortcomings in our study, from the perspective of methods, interpretations, study design as well as dataset and variable choices.

**Reviewer #1 Technical Comment 24:**

Technical comments:
I really like that they provided an appendix with acronyms and I used it a lot when I was reviewing the paper. But there were a few things missing.
- Figure 1. Since the authors have an appendix for acronyms, I suggest they add the definitions of the abbreviations for the individual predictors there as well, and reference that in the figure caption.

**Authors' Response:**

Thank you for this comment. We have added all the abbreviations of the individual predictors in the Acronyms section and referenced them in the caption of Figure 1.

**Reviewer #1 Technical Comment 25:**

- R22 is not defined in the text or in the appendix. Presumably it is Roychoudhury et al., 2022?

**Authors' Response:**

Thank you for this comment. We apologize for this mistake and have included a post-note in the citation of Roychoudhury et al., 2022 to mention R22 and also in the abbreviations list in Appendix B.

**Reviewer #1 Technical Comment 26:**

Please add RGI to the abbreviations list.

Thank you for this comment. We have added RGI to the abbreviations list in Appendix B.

**Reviewer #1 Technical Comment 27:**

L44: Did the authors mean, "and potentially leading to the misattribution of the relevant drivers to snowmelt"?

**Authors' Response:**

Thank you for this comment. We understand that the previous statement made was too assertive

and we modified the text to mention that that it can be a potential misattribution, shown as follows,

> *The spatial heterogeneity in HMA's glaciers and the non-linear interaction between these processes can either intensify or buffer the response of the climate system, confounding the net effect of AMI on the cryosphere and potentially leading to the misattribution of the relevant drivers to snowmelt (Refs.).*

**Reviewer #1 Technical Comment 28:**

L. 63: Two sentences right after each other with lists seems overwhelming. Consider re-writing.

**Authors' Response:**

Thank you for this comment. We have modified these two sentences to allow for a more coherent flow, shown as follows,

> *Challenges in constraining these non-linear interactions in these models arise from a variety of factors: the sparsity of long-term continuous observations, theoretical uncertainties in parameterization and coupling, spatio-temporal heterogeneity in the processes, as well as the magnitude and direction of these feedbacks. Researchers have traditionally addressed these uncertainties through multiple approaches: 1) perturbing model parameters and sensitivity analysis; 2) selective withholding of modeled parameters and comparing their relative impacts; 3) using observations and data assimilation (Schneider et al., 2017), and 4) assessing emergent constraints across multiple models, (Heinze et al., 2019; Soden et al., 2008; Zhou et al., 2019; Moch et al., 2022; Gettelman, 2015; Barthlott et al., 2022; Archer-Nicholls et al., 2016; Usha et al., 2020; Stein and Alpert, 1993).*

**Reviewer #1 Technical Comment 29:**

The intro seemed to ramble a bit. There is an opportunity to improve the paper by getting to the point more succinctly in this section.

**Authors' Response:**

Thank you for this feedback. We have substantially revised the introduction to be more concise, reducing its length while maintaining a coherent structure. The modified introduction establishes the research problem more directly in the opening paragraphs, eliminates redundant explanations of frameworks and methods, and presents our research objectives and methods more clearly. We have also added an outline of the manuscript for clarity.

**Reviewer #1 Technical Comment 30:**

Equation 1: why is $Y_{SCF}^{s,t}$ used instead of $Y^{s,t}$? on Line 213, $Y$ is defined as SCF. And just $Y$ is used in equations 2 and 3.

**Authors' Response:**

Thank you for this comment. We have now replaced $Y_{SCF}$ with $Y$ in equation 1 to maintain similarity with other mentions of $Y$ in the text.

**Reviewer #1 Technical Comment 31:**

Equation 1: Please indicate what the double O in term 3 means/represents.

**Authors' Response:**

Thank you for this comment. The double O ($O$) in Term 3 of Equation 1 refers to higher-order terms based on the predictors that are not considered in Terms 1 and 2. These can include functional representations of multiple predictors together (more than two different predictors), while Term 2 only considers two distinct predictors.

**Reviewer #1 Technical Comment 32:**

L240: also reference supplemental Table 2.

**Authors' Response:**

Thank you for this comment. We have mentioned Supplementary Table 2 in Line 240.

**Reviewer #1 Technical Comment 33:**

Table 1: please define mu and sigma in the Table caption, and not just in the text.

**Authors' Response:**

Thank you for this comment. We have included the definitions of $\mu$ and $\sigma$ in the caption of Table 1.

**Reviewer #1 Technical Comment 34:**

Figure 3. Please describe what "Others" means in the caption (or just replace "Others" with "AOD550+Sea salt")

**Authors' Response:**

Thank you for this comment. We have defined Others in the caption of Figure 3 for clarification.

Figure 3. In the figure, the word "contribution" is too vague. Please clarify what this means in the caption or use more specific wording in the figure.

**Authors' Response:**

Thank you for this comment. The caption has been modified to clarify "contribution" as follows,

*Flow diagrams depicting the sum of the contribution of four aerosol groups and five meteorology groups to AMI in low snow-cover regions during the late snowmelt period (May-July) across the Obs-Model (a) and Model-Model construct (b), similar to the donut plots in Fig. 2(a) and 2(d), except the contribution is aggregated for all three reanalyses and both importance metrics. This contribution (also expressed in %) is the importance values ($\alpha$ in %) of the AER and MET predictors normalized to the total mean importance of AMI on snow. The top row shows the contributions color-coded by aerosol groups, and the bottom row shows the same contributions color-coded by meteorology groups. The Others aerosol group refers to the combined importance of both AOD at 550 and sea salt surface mixing ratio. The flow diagrams are made using SankeyMATIC.*

**Response to Reviewer #2**

**Reviewer #2 General Comment 1:**

The authors make use of statistical and machine learning methods applied to satellite observations and reanalyses data to assess the importance of various (non-linear) links among multiple aerosol-meteorology-snow variables over high-mountain Asia during the late snowmelt season (May-July). This information is then used to infer details on the skilful representation of related physical processes and make recommendations for ESMs development. The approach is interesting and reveals some important relationships between variables. From this point of view, the analysis is systematic and overall comprehensive by targeting an extensive set of variables. The manuscript is well written and the figures are well done. Yet, I would recommend making the text more concise and substantially shorter.

**Authors' Response:**

Thank you for the comments. In response to the suggestion for a more concise manuscript, we have substantially revised the introduction to be more concise, reducing its length while maintaining a coherent structure. The modified introduction establishes the research problem more directly in the opening paragraphs, eliminates redundant explanations of frameworks and methods, and presents our research objectives and methods more clearly.

**Reviewer #2 Main Comment 2:**

My main comments are related to the actual insights into the physical mechanisms the analysis can identify. While the relationships found are undoubtedly important, it is not really clear to me how they can help to identify physical pathways, especially causal links, associated for example to seasonal/interannual/decadal variability of snow cover over HMA.

**Authors' Response:**

Thank you for this important point about physical pathways and causal links. We agree that our findings do not directly identify causal links or physical mechanisms between the aerosol, meteorology variables, and snow cover variability. We have modified our objective (in the last paragraph of the introduction) and the Limitations and Future Directions (Section 4.3) to reflect this. The main aspects of the revisions are as follows,

- **Revised objective:** We have modified our objective in the last paragraph of the introduction to reflect that the primary aim of our study is not to establish causality or point towards physical mechanisms, but rather to identify and quantify the relative importance of variable interactions that significantly influence snow cover fraction processes.

  *The aim is to gain insights into the model representation of AMI-related processes in each reanalysis, rather than discovering new physical mechanisms or causal pathways*

*within AMI.* (Last paragraph of the introduction.)

- **Revised title:** We have revised our title to *"Diagnosing Aerosol-Meteorological Interactions on Snow within Earth System Models: A Proof-of-Concept Study over High Mountain Asia"* to emphasize that our study focuses on diagnosing the differences in the representation of interactions relevant to snowmelt across three reanalyses (and the inherent model design) rather than identifying observable interactions or physical pathways.

- **Revised limitations section:** We have modified our limitations section to reflect that our approach is purely statistical and that identifying statistical relationships is distinct from establishing causality. We have discussed in our Limitations and Future Directions (Section 4.3) how our findings reflect correlative patterns and can only hint towards causal links. We also mention how our results should inform subsequent process-based modeling studies to investigate causality and physical pathways. Our modification is as follows,

  > *This potential for misattribution is an inherent limitation in our diagnostic framework and underscores the importance of interpreting our results as indicative of correlative relationships, rather than definitive causal links.*

  > *We recognize that our approach is statistical in nature and does not directly identify physical mechanisms. While we can identify significant relationships and interactions among variables, establishing definitive causality requires detailed, process-based investigations. As such, our findings should be interpreted as highlighting potential relationships that can inform and guide subsequent detailed physics-based modeling efforts.*

  We also mention that this approach is more of a proof-of-concept study that can provide insights into which interactions are most relevant and how their representation differs across reanalyses. We only hint towards causality and physical mechanisms from established literature to support our results. Our modifications in the Limitations and Future Directions (Section 4.3) are reflected as follows,

  > *The value lies in 1) identifying biases across different models in representing non-linear interactions, which is crucial for investing in more computationally expensive process-based modeling; and 2) bridging the gap between process understanding and model evaluation by quantifying specific biases in the representation of interactions, beyond estimating general biases in the model outputs.*

- **Clarified approach:** To aid in interpreting our findings as to which interactions impact snow cover, we have added to our methodology section to explain how our variable groupings conceptually align with modules within ESMs to aid in interpretation of our results, as follows,

*The variable groupings here are meant to reflect the modules within the architecture of coupled models/ESMs that interact as sub-components, aligning with a systems-science framework.*

- While our findings do not identify new physics pathways, we have revised our manuscript to clearly support our inferences with previous studies that have identified relevant physical pathways and causal links. Section 3.3.5 (Bringing it altogether) highlights some of these established mechanisms, some of which we show below. Section 3.4 also analyzes joint distributions using observations of the predictors across the reanalyses to identify similarities with established mechanisms.

*The underrepresented dust-circulation interactions from our network analysis align directly with the physical mechanisms described by (Lau and Kim, 2018) regarding the snow-monsoon relationship in Asia. Their modeling experiments showed that the deposition of dust on snow initiates a series of interconnected processes: reduced snow albedo, increased solar radiation absorption, and accelerated snowmelt, with subsequent modification of regional circulation patterns. Specifically, they showed that dust deposition in April-June leads to atmospheric warming and pressure patterns (changes in geopotential height) that enhance dust transport to the Himalayan-Indo-Gangetic region. Our network diagrams reveal that these critical connections between dust and circulation variables (particularly geopotential height and mean sea level pressure) are insufficiently captured across all three reanalyses, despite being important for SCF variability. This explains persistent biases in SCF and dust, particularly in LSC regions nearest to major dust sources like the Taklamakan Desert (see Supplementary Fig. 1 and 4), which Lau and Kim identified as contributing significantly to dust deposition on Himalayan snow. (Zhao et al., 2024) confirms the role of dust in impacting the Asian summer monsoon and how more accurate dust simulations can help constrain the monsoon circulation patterns. The progression in interaction complexity we observe from ERA5/CAMS4 to MATCHA shows improvement but still indicates insufficiencies in representing the dust-snow-circulation feedbacks that are crucial for regional climate dynamics.*

**Reviewer #2 Main Comment 3:**

I feel the work described here is largely statistical, quite dry, and offers limited information on how the relationships found can be applied to understand snow variability. Ultimately, this is what is partially misrepresented in current ESMs, and this would be the important value-added towards discerning relationships that can improve projections of water availability.

**Authors' Response:**

Thank you for the candid assessment. We acknowledge the limitations of statistical approaches and have revised our manuscript to better articulate the value of our methodology, as follows,

- **Bridge statistical inference and process understanding:** While our approach is indeed statistical, it serves as a diagnostic tool to identify which interactions are poorly represented across reanalyses. Rather than providing new physical insights, we systematically quantify biases in how different modeling frameworks represent key interactions affecting snow cover, which can guide more targeted process-based investigations. We modified this in the Limitations and Future Directions (Section 4.3) as mentioned previously.

- **Guiding model development:** By identifying specific interaction deficiencies across different reanalyses (particularly dust-circulation interactions), our approach pinpoints where ESM development efforts should focus to improve snow cover prediction. This addresses a fundamental challenge in model development, knowing which processes to prioritize for improvement among myriad possibilities. We modified this in the Limitations and Future Directions (Section 4.3) as mentioned previously.

- **Addressing complexity:** Traditional approaches often examine either meteorological drivers or aerosol impacts in isolation. Our approach simultaneously evaluates non-linear interactions across land, meteorology, and chemistry interfaces, revealing potential misattributions in both prediction systems and attribution studies. We have revised the second paragraph of our Introduction to explicitly mention this. We added this sentence while discussing previous studies,

  *Previous studies rarely incorporate this full spectrum of variables and pathways in hydrological analyses*

- **Practical utility:** We have revised the Limitations and Future Directions (Section 4.3) to reflect the takeaway from our study, as follows,

  *As such, our findings should be interpreted as highlighting potential relationships that can inform and guide subsequent detailed physics-based modeling efforts. As reanalyses and ESMs continue to evolve, our approach can support these development efforts by pinpointing the most critical coupling processes. Given the proof-of-concept nature of our study, applying similar analyses to other regions or phenomena could yield equally*

*valuable insights into model representation of Earth system interactions.*

We view our approach as complementary to process-based studies. We identify where models disagree on important interactions related to snowmelt, to narrow the focus for subsequent mechanistic investigations that can ultimately improve projections of water availability in this climate-vulnerable region.

**Reviewer #2 Main Comment 4:**

As such, the scope of the manuscript is a bit limited and I think can lead to misattribution of cause and effect. Additionally, I would argue that some links may be compounded by other factors, and may not be really separable. For example, the link with circulation (around L505-510) seems quite obvious. It would be more useful, for example, to identify patterns of upper-level circulation. Similarly, the paragraph starting at L598 is inconclusive and quite speculative.

**Authors' Response:**

Thank you for these thoughtful comments regarding the misattribution and compounding effects. We have carefully revised our manuscript to address these concerns as follows,

- **Compounding effects:** Regarding the compounding effects, we agree that many of these interactions are not clearly separable and can be misinterpreted. We have mentioned in the introduction, Section 3.3.4 and Section 4.3, the concept of "buffering" and discussed how our network-based approach can help visualize these relationships. We have also highlighted how our approach cannot determine the compounding effects as of now but tried to confirm the validity of our results through previous studies and our use of model constructs. The text is shown as follows,

  *The spatial heterogeneity in HMA's glaciers and the non-linear interaction between these processes can either intensify or buffer the response of the climate system, confounding the net effect of AMI on the cryosphere and potentially leading to the misattribution of the relevant drivers to snowmelt.* (Introduction, paragraph 2.)

  *An associated issue with misattribution is the buffering of the snowmelt response from one predictor due to the presence of other predictors which can obscure the true influence of especially the aerosol predictors on snowmelt, resulting in inaccurate conclusions about their relative contributions.* (Potential Misattributions in each Reanalysis, section 3.3.4).

  *Although we can potentially highlight where each model misattributes the snowmelt sensitivity for AMI interaction, we are unable to determine the buffering of snowmelt response of AER predictors by the MET variables with the current approach.* (Potential Misattributions in each Reanalysis, section 3.3.4).

*An additional avenue to explore is the separation of misattribution and buffering among the drivers in the identified couplings, which is limited in our current approach. Specifically, our approach does not fully separate compounding or buffering effects where multiple drivers impact simultaneously.* (Limitations and Future Directions, Section 4.3)

- **Circulation patterns (L505-510):** We agree that identifying specific upper-level circulation patterns would be valuable. While a detailed analysis of circulation patterns is beyond our current scope, we've clarified this limitation and suggested it as a direction for future process-based studies in Section 4.3, as follows,

  *Furthermore, identifying specific upper-level circulation patterns that drive SCF variability could builds upon our findings for deeper insights. While our approach identifies the importance of circulation-related variables and their interactions across different reanalyses, a detailed assessment of circulation regimes would complement our statistical analyses with a more process-based understanding.*

- We have revised the paragraph starting at L598 to more clearly distinguish between our findings and their potential implications and focused on the concrete differences we observed between reanalysis products, as follows,

  *Circulation-related interactions with dust aerosols, particularly those involving geopotential height and mean sea level pressure, are found to be significant yet insufficiently represented in the models within each reanalysis. Previous studies have mentioned the uncertainty with dust and circulation across models and how the interactions between the two initiate feedbacks affecting monsoon and snowmelt in HMA (discussed in Sect 3.3.5). The importance of circulation-related interactions suggests that interactions of absorbing aerosols and smaller sub-grid processes with large-scale atmospheric circulation involving clouds, convection, and transport across the boundary layer need to be addressed for more accurate snow hydrology and understanding of Asian monsoon dynamics.* (Main Findings, Section 4.1)

We have been careful to distinguish between our empirical findings and their potential implications, focusing primarily on quantifying differences in how various reanalyses represent key interactions rather than making definitive causal inferences about underlying physical mechanisms.

---

## Author Response (AR2)

**Response for "Diagnosing Aerosol-Meteorological Interactions on Snow within Earth System Models: A Proof-of-Concept Study over High Mountain Asia"**

**Response to Reviewer #1**

**Reviewer #1 Minor Comment 1:**

The authors have very nicely addressed most of my points in the response to reviewers. I only have a very minor comment remaining.

The response to Reviewer 1 Specific Comment 11 was useful as explained to me, but it would be more useful if the authors clarified this information in the text for the benefit of the other readers. For example, they could say (assuming this is correct): "where the dependence ($\alpha$) on a predictor $X_i$ is not a constant, but dependent on a second predictor $X_j$ (what we call a pairwise interaction contribution)."

and

"Note that the metrics of importance are bivariate, reflecting the joint sensitivity of SCF to a predictor in the presence of another predictor (the pairwise interaction contribution). Importance of AMI on snow can thus be interpreted as the impact of MET predictors on snow in the presence of AER variables."

and "non-linear interaction terms defined as product terms between these predictors (253 in total, see section 2.6)"

and in section 2.6, perhaps some of what is described here to me could be included in the text for readers trying to understand the methods. I just feel like this will help with clarity.

**Authors' Response:**

Thank you for the comment. We have modified some parts of the text, especially in Sections 2.5, 2.6, and 2.7, to clarify more on the methodology and the metrics.

In Section 2.5, we modified the paragraph 2 as follows:

> *The importance metric, $\alpha$, is derived with two distinct methods: (1) relative importance (RI), obtained from the multi-linear regression described in Sect. 2.6 includes the linear predictors and their second-order product terms (Terms 1 and 2 in Eq. 1). The estimated importance values ($\alpha$) are in percentages, and the sum for all terms in the regression equals 100%. (2) Shapley contribution (SHAPc) calculated from an ML model introduced in Sect. 2.7. It is important to note that while the multiple linear regression for RI is trained on both the original predictors and the product terms to account for interacting effects, the ML model is trained only on the individual predictors, as its built-in feature contribution algorithm (see Sect. 2.7) also accounts for the pairwise interactions, acting as a bulk measure of the importance, thus the three terms in Eq. 1 (Term 1 to 3). The importance values calculated from machine learning*

*are normalized so that their total also equals 100%. Thus, both importance metrics are expressed as percentages that sum to 100%, making their magnitudes directly comparable. Each $\alpha$ value is inherently bivariate as it quantifies the sensitivity of snow cover fraction (SCF) to a given predictor in the presence of another predictor. Importance of AMI on snow can thus be interpreted as the impact of MET predictors on SCF in the presence of AER variables.*

In Section 2.6, we modified the text as follows for clarity:

*where $N(= 22)$ is the original number of predictors (see Fig. 1 and Supplementary Table 2 for the 22 predictors: six aerosol, 15 meteorological, and an elevation variable) representing the main effects, in addition to $(\binom{N}{2} = 231)$ non-linear interaction terms defined as product terms between these predictors (excluding square terms), thus leading to 253 (= 231 + 22) predictors in total. We explicitly define second-degree interaction terms in the MLR model (only non-square terms) shown in Eq. (??) to represent the non-linear sensitivities of our predictors to the SCF variability for each GR and each month in the late snowmelt season. The interaction terms belong to five groups, namely: 1) AER-AER, 2) AER-MET, 3) AER-ELEV, 4) MET-ELEV, and 5) MET-MET. Eq. (??) offers us an alternate understanding of such a pairwise interaction, where the dependence ($\alpha$) on a predictor $X_i$ is not a constant, but dependent on a second predictor ($X_j$).*

In addition, the Supplementary Text S1 details the implementation of the SHAPc contribution (see Section S1.4) from the raw SHAP values from the XGBoost model. We mentioned this in Section 2.7 as follows,

*The SHAP values were normalized to percentages, defined hereafter as SHAPc, by averaging the absolute SHAP values and dividing by their sum. This enables an analogous comparison to the RI metric as a percentage contribution to the total SCF (target) response. Additional details on this implementation are available in the Supplementary Information (Sect. S1.4).*